# Cold-inducible RNA-binding protein (CIRBP) adjusts clock-gene expression and REM-sleep recovery following sleep deprivation

Marieke MB Hoekstra, Yann Emmenegger, Jeffrey Hubbard, Paul Franken*

Center for Integrative Genomics, University of Lausanne, Lausanne, Switzerland

**Abstract** Sleep depriving mice affects clock-gene expression, suggesting that these genes contribute to sleep homeostasis. The mechanisms linking extended wakefulness to clock-gene expression are, however, not well understood. We propose CIRBP to play a role because its rhythmic expression is i) sleep-wake driven and ii) necessary for high-amplitude clock-gene expression *in vitro*. We therefore expect *Cirbp* knock-out (KO) mice to exhibit attenuated sleep-deprivation-induced changes in clock-gene expression, and consequently to differ in their sleep homeostatic regulation. Lack of CIRBP indeed blunted the sleep-deprivation incurred changes in cortical expression of *Nr1d1*, whereas it amplified the changes in *Per2* and *Clock*. Concerning sleep homeostasis, KO mice accrued only half the extra REM sleep wild-type (WT) littermates obtained during recovery. Unexpectedly, KO mice were more active during lights-off which was accompanied with faster theta oscillations compared to WT mice. Thus, CIRBP adjusts cortical clock-gene expression after sleep deprivation and expedites REM-sleep recovery.
DOI: https://doi.org/10.7554/eLife.43400.001

*For correspondence:
paul.franken@unil.ch

## Introduction

The sleep-wake distribution is coordinated by the interaction of a circadian and a sleep homeostatic process (*Daan et al., 1984*). The molecular basis of the circadian process consists of clock genes that interact through transcriptional/translational feedback loops. CLOCK/NPAS2:BMAL1 (ARNTL) heterodimers drive the transcription of many target genes, among them the *Period* (*Per1-2*), *Cryptochome* (*Cry1*, −*2*) and *Rev-Erb* (*Nr1d1*, −*2*) genes. Subsequently, PER:CRY complexes inhibit CLOCK/NPAS2:ARNTL-transcriptional activity thereby preventing their own transcription. In addition, clock components such as the transcriptional repressor NR1D1 regulate the transcription of *Arntl*, ensuring together with other transcriptional feedback loops a period of ca. 24 hr (*Lowrey and Takahashi, 2011*). While this clock-gene circuitry is functionally expressed in almost each cell of the body, peripherally generated circadian rhythms are coordinated by a central pacemaker in the suprachiasmatic nuclei (SCN) of the hypothalamus, assuring proper circadian time-keeping at the organismal level (*Hastings et al., 2018*).

The sleep homeostatic process keeps track of time spent awake and asleep, during which sleep pressure is increasing and decreasing. The mechanisms and specific brain structure(s) underlying this process are to date unknown. Accumulating evidence has implicated clock genes in sleep homeostasis (reviewed in *Franken, 2013*). This is supported by studies in several species (i.e. mice, fruit flies and humans), showing that mutations in circadian clock genes are associated with an altered sleep homeostatic response to sleep deprivation (e.g. *Mang et al., 2016*; *Shaw et al., 2002*; *Viola et al., 2007*; *Wisor et al., 2002*; *He et al., 2009*). Sleep homeostasis has distinct local, use-dependent aspects (*Krueger and Tononi, 2011*), a notion which seems incompatible with the existence of a

central sleep homeostat, functionally analogous to the SCN as a master clock in circadian timekeeping. Disrupting circadian rhythms in the tuberomammillary nucleus, however, through removal of the core clock gene *Arntl*, affects the regulation of time spent in NREM sleep after sleep deprivation, implying that specific nuclei can impact aspects of the sleep homeostatic process (*Yu et al., 2014*). An additional argument for a role for clock genes in sleep homeostasis comes from studies showing that enforced wakefulness affects the expression of clock genes, such as *Nr1d1*, *Per1-3* and *Dbp*, in the brain (*Mongrain et al., 2010*). These sleep-deprivation induced changes in clock-gene expression were most pronounced in the cerebral cortex, while expression in the SCN remained unperturbed (*Curie et al., 2015*). The mechanisms through which these changes occur are, however, unclear.

In this study, we examined one possible mechanism and hypothesized that some of the sleep deprivation-induced changes in clock-gene expression occur through Cold-Inducible RNA Binding Protein (CIRBP). Decreasing temperature *in vitro* increase CIRBP levels (*Nishiyama et al., 1997*) and daily changes in mice core body temperature are sufficient to drive robust cyclic levels of *Cirbp* and CIRBP (*Morf et al., 2012*) in anti-phase with temperature. Although daily changes in cortical temperature appear circadian, in the rat more than 80% of its variance is explained by the sleep-wake distribution (*Franken et al., 1992*). Hence, when controlling for the daily sleep-wake driven changes in cortical temperature by sleep deprivations, the daily rhythms of cortical *Cirbp* become strongly

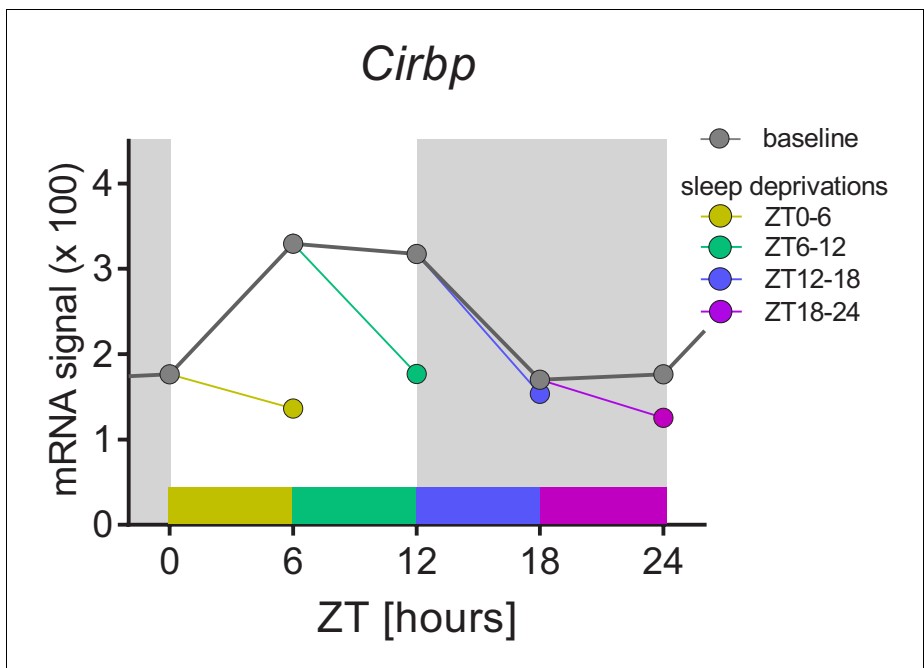

**Figure 1.** The sleep-wake distribution drives daily changes of *Cirbp* expression in the mouse brain. Dark-grey symbols and line (baseline): from ZT0 to ZT12, mice spend most of their time asleep and *Cirbp* increases, whereas between ZT12-18, when mice spent most of their time awake, *Cirbp* decreases. When controlling for the daily occurrence in sleep by performing four 6 hr sleep deprivation starting at either ZT0, −6,–12, or −18 (each sleep deprivation is annotated with its own color), the diurnal amplitude of *Cirbp* is greatly reduced (colored circles represent level of *Cirbp* expression reached at the end of each sleep deprivation). Nine biological replicates per time point and condition were used (one data point missing at ZT18), and RNA was extracted from whole brain tissue (see *Maret et al., 2007* for details). Data were taken from GEO GSE9442. Light-grey areas represent the dark periods.

DOI: https://doi.org/10.7554/eLife.43400.002

The following source data is available for figure 1:

**Source data 1.** Expression of Cirbp in the brain under undisturbed conditions and after controlling for the circadian distribution of sleeping and waking.

DOI: https://doi.org/10.7554/eLife.43400.003

attenuated (see *Figure 1*, based on Gene Expression Omnibus number GSE9442 from *Maret et al., 2007*). Furthermore, the expression of the gene *Cirbp* shows the highest down-regulation of all genes after sleep deprivation (*Mongrain et al., 2010*; *Wang et al., 2010*), underscoring its sleep-wake-dependent expression. But how does CIRBP relate to clock gene expression?

Two independent studies have shown that the temperature-driven changes in CIRBP are required for high amplitude clock-gene expression in temperature-synchronized cells (*Morf et al., 2012*; *Liu et al., 2013*). Therefore, we and others (*Archer et al., 2014*) proposed that changes in clock-gene expression during sleep deprivation are a consequence of the sleep-wake-driven changes in CIRBP. To test this hypothesis, we first quantified in mice the contribution of sleep-wake state and locomotor activity to changes in cortical temperature. Next, we measured sleep-deprivation-induced changes in clock-gene expression in both (wild-type) WT and mice lacking CIRBP (*Cirbp* KO) (*Masuda et al., 2012*), anticipating that clock-gene expression in response to sleep deprivation differed in KO mice. Finally, as clock genes play a role in sleep homeostasis (*Franken, 2013*), we also compared the homeostatic regulation of sleep between WT and KO mice.

Our results demonstrate that like in the rat, the sleep-wake distribution in the mouse is the major determinant of cortical temperature changes, with a significant, albeit small, contribution of locomotor activity. In line with our predictions, we found that a lack of CIRBP attenuated the sleep-deprivation-induced changes in the cortical expression of *Nr1d1* and the homeostatic response in REM-sleep time. However, in contrast to our hypothesis, we observed that differences in *Per2* and *Clock* expression after sleep deprivation were augmented in *Cirbp* KO mice. Unexpectedly, these mice were also substantially more active during the dark phase of the 24 hr period when compared to their WT littermates, without increasing their time spent awake. This increase in locomotor activity was accompanied by an acceleration of electroencephalogram (EEG) theta oscillations during active waking. Altogether, our data show that *Cirbp* contributes to some of the sleep-deprivation-induced changes in clock-gene expression, but also points to the existence of other sleep-wake-driven pathways transferring sleep-wake state information to clock-gene expression.

## Results

### The relation between cortical temperature, sleep-wake distribution, and locomotor activity

The dependence of brain and cortical temperature on sleep-wake state has been demonstrated in a number of mammals (*Alföldi et al., 1990*; *Baker and Hayward, 1968*; *Deboer et al., 1994*; *Franken et al., 1992*; *Hayward and Baker, 1968*) but has not been specifically addressed in the mouse. Moreover, none of these studies controlled for locomotor activity when quantifying the contribution of sleep-wake state to brain temperature. We therefore measured cortical temperature, locomotor activity and sleep-wake state in WT and *Cirbp* KO mice during two baseline days, a 6 hr sleep deprivation and the following 2 recovery days. Because the relationship between cortical temperature, locomotor activity and waking in WT and KO mice was very similar, most of the results are illustrated for WT mice only.

### Fast changes in cortical temperature occur at sleep-wake state transitions

A representative example of a 96 hr recording of locomotor activity, sleep-wake state and cortical temperature is depicted in *Figure 2*. Consistent with mice being nocturnal animals, increased waking, locomotor activity and overall higher cortical temperature were observed during the dark phase. Sleep deprivation [between Zeitgeber Time (ZT)0–6] led to an almost uninterrupted period of 6 hr waking, during which locomotor activity and cortical temperature reached values comparable to those reached during bouts of spontaneous wakefulness under undisturbed baseline conditions (i.e. ZT12-18). Closer inspection of cortical temperature changes revealed rapid fluctuations associated with sleep-wake state transitions. We quantified these changes in cortical temperature by selecting and aligning transitions between consolidated bouts of NREM and REM sleep and wakefulness during the 2 baseline days (*Figure 2*-B). When entering NREM sleep, cortical temperature consistently decreased, whereas at transitions into wake and REM sleep, it increased. This latter transition was characterized by a fast and consistent change in cortical temperature, where within 1.5 min an

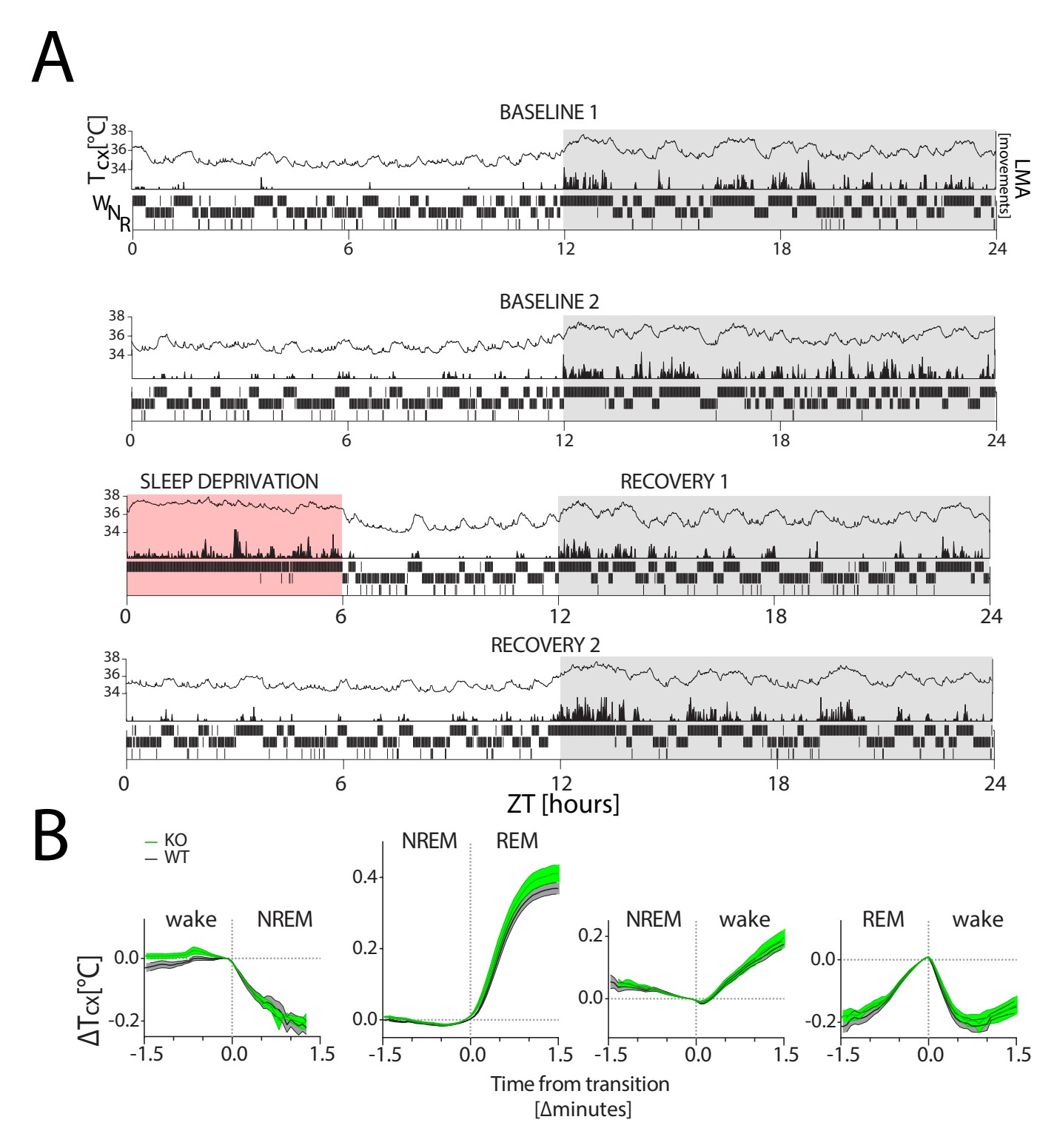

**Figure 2.** Cortical temperature changes with sleep-wake state. $T_{cx}$: cortical temperature; LMA: locomotor activity. (**A**) A representative 4-day recording of one mouse in LD 12:12 (in white:grey) during 2 baseline days (top two panels), followed by a 6 hr SD (in red; third panel) and 2 recovery days (bottom two panels), with within each panel $T_{cx}$ (top; line graph), LMA (middle; area plot) and sleep-wake states (bottom; hypnogram). Sleep-wake states are averaged per minute to aid visualization. LMA was collected and plotted per minute (see Methods). (**B**) Changes in $T_{cx}$, depicted as mean ± SEM (WT: black lines, grey areas; KO: green lines and areas) relative to $T_{cx}$ at the sleep-wake transition (average of the last value before and first value after transition). $T_{cx}$ increased when transitioning from NREM sleep to wake and to REM sleep (two-way RM ANOVA, factors genotype (GT) and Time, factor

*Figure 2 continued on next page*

*Figure 2 continued*

Time, F(38,418)=126, F(45,540)=535.5; p<0.0001, respectively) and decreased when transitioning from wake to NREM sleep (F(22, 242)=1661, p<0.0001). Also the transition from REM sleep to wake affected the time course of $T_{cx}$ (F(23,276)=131.8, p<0.0001). GT interacted with time during the wake to NREM sleep transition (F(22,242)=1.8, p=0.01), but not for other sleep-wake state transitions (p>0.12). No significant GT effect was detected (p>0.12) Transition data were obtained from both baseline days (see Materials and methods for details).

DOI: https://doi.org/10.7554/eLife.43400.004

The following source data is available for figure 2:

**Source data 1.** Example of a 4-day recording; the transitions in sleep-wake state and associated changes in cortical temperature in Cirbp WT and KO mice.

DOI: https://doi.org/10.7554/eLife.43400.005

increase of 0.4°C was observed. Transitions from REM sleep into wake led to an initial decrease in cortical temperature, contrasting with the increase observed in the wakefulness when following NREM sleep. Altogether, these results provide evidence that sleep-wake state importantly contributes to changes in cortical temperature.

## Daily cycles in cortical temperature are driven by sleep-wake state

After having established that rapid changes in cortical temperature were indeed evoked by changes in sleep-wake state, we next sought to determine whether the significant changes in cortical temperature across the day were also due to overall rhythms in sleep-wake state, and potentially locomotor activity. Examining cortical temperature, waking and locomotor activity per hour showed an oscillation over the course of the 24 hr baseline period (*Figure 3*-A; two-way RM ANOVA on averaged 24 1 hr intervals during baseline; Factor Time: F(23,207), cortical temperature, F = 70.5, waking: F = 27.2, locomotor activity: F = 22.5; p<0.0001). The time course of cortical temperature did not differ between genotypes (*Figure 3—figure supplement 1*), neither did the amplitude of the baseline change (WT: 2.34 ± 0.1, KO: 2.33 ± 0.1; t-test: t(9)=-0.02, p=0.98; average of the difference between the highest and lowest hourly value in baseline days 1 and 2). Importantly, the time course of cortical temperature was strongly correlated to both waking (*Figure 3*-B left: WT: $R^2$ = 0.76; KO: $R^2$ = 0.81, p<0.0001) and locomotor activity (*Figure 3*-B right: WT: $R^2$ = 0.60; KO: $R^2$ = 0.72, p<0.0001).

To assess the influence of waking on cortical temperature further, mice were sleep deprived between ZT0 and ZT6. During the 6 hr sleep deprivation, animals spent 98% awake of the 6 hr, significantly more when compared to the same circadian time under baseline conditions (paired t-test: waking [hours] baseline: 2.2 ± 0.1, sleep deprivation: 5.9 ± 0.04; t(10)=-38.1, p<0.0001; $\log_2$[movements], baseline: 13.1 ± 2.4, sleep deprivation: 39.4 ± 2.1; t(10)=-15.2, p<0.0001). This increase in waking and activity led to sustained elevated cortical temperatures (average ZT0-ZT6 [°C]: baseline: 34.7 ± 0.07, sleep deprivation: 36.6 ± 0.06, t(10)=-44.3, p<0.0001), suggesting a causal relationship. Notably, the genotype of the mice did not contribute to or interact with these changes (two-way ANOVA, Genotype*sleep deprivation/baseline: p>0.39)

However, factors other than extended waking consequent to sleep deprivation, such as stress, could have contributed to these changes in cortical temperature. To address this issue, we selected in each mouse the longest uninterrupted spontaneous waking bout occurring during baseline (average length: 100 ± 19 min). We then compared cortical temperature of this bout with values reached of an equivalent time spent awake from the start of the sleep deprivation. The average of the last 10 min of these bouts was taken to reduce the influence of temperature differences at bout-onset. Cortical temperature reached in these bouts did not differ during sleep deprivation and spontaneous wakefulness in WT mice (*Figure 3*-C), nor in KO mice (t(5)=0.84, p=0.44), indicating that factors other than extended wakefulness (e.g. light exposure, circadian time, sleep deprivation-associated stress) do not expressly contribute to these changes.

Considering the strong correlation between locomotor activity and cortical temperature (WT: $R^2$ = 0.76; KO: $R^2$ = 0.81; p<0.0001), it could be hypothesized that locomotor activity contributes to the sleep-wake associated changes in cortical temperature. To investigate this further, the respective contribution of waking and locomotor activity to changes in cortical temperature was quantified by partial correlation analysis. Although locomotor activity did significantly contribute, substantially

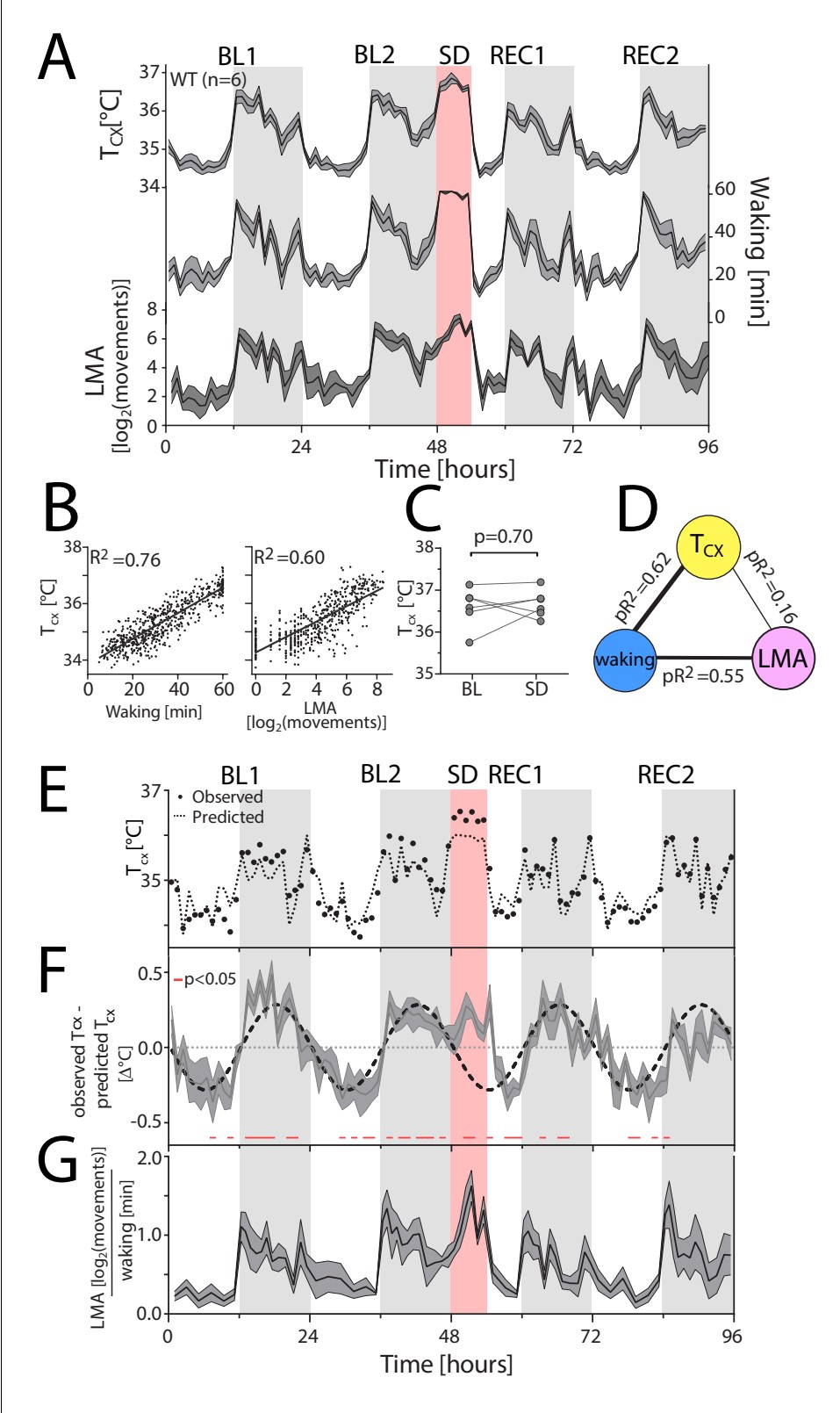

**Figure 3.** Waking is the major determinant of cortical temperature. For panels A, F and G, the dark-grey line represents the mean, grey areas span ± 1 SEM. BL: baseline, SD: sleep deprivation, REC: recovery, $T_{cx}$: cortical temperature, LMA: locomotor activity. Light-grey areas mark the dark periods. (**A**) Time course of hourly values of $T_{cx}$, waking and LMA across the entire experiment. (**B**) Both waking (left) and LMA (right panel) strongly correlated with $T_{cx}$ (n = 6; 96 values per mouse; p<0.0001). $R^2$: correlation coefficients. (**C**) $T_{cx}$ during SD did not differ from levels reached after long waking bouts

*Figure 3 continued on next page*

*Figure 3 continued*

during BL (t(5)=0.41, p=0.70). (**D**) Waking after correcting for LMA is the major determinant of $T_{cx}$, as revealed by partial correlation analysis; here performed on the combined hourly values of all WT mice. $pR^2$: partial correlation coefficients. (**E**) A representative example [mouse TC03], with measured $T_{cx}$ (closed circles), and predicted $T_{cx}$ (stippled line) based on the correlation between $T_{cx}$ and waking. (**F**) During the dark phase and SD, the predicted $T_{cx}$ is lower than the measured $T_{cx}$, resulting in positive residuals [residuals: observed $T_{cx}$ – predicted $T_{cx}$], whereas during the light phase, the predicted $T_{cx}$ is higher than the measured $T_{cx}$, resulting in negative residuals (t-test: data <> 0, p<0.05, red lines underneath the curves). (**G**) LMA per unit of waking follows a similar pattern as the residuals in Panel F.

DOI: https://doi.org/10.7554/eLife.43400.006

The following source data and figure supplements are available for figure 3:

**Source data 1.** Waking, LMA and cortical temperature.
DOI: https://doi.org/10.7554/eLife.43400.011
**Figure supplement 1.** Cortical temperature ($T_{cx}$) shows similar daily variation in KO and WT mice.
DOI: https://doi.org/10.7554/eLife.43400.007
**Figure supplement 1—source data 1.** Cortical temperature during baseline in Cirbp WT and KO mice.
DOI: https://doi.org/10.7554/eLife.43400.008
**Figure supplement 2.** The residuals of the optimized mixed linear model.
DOI: https://doi.org/10.7554/eLife.43400.009
**Figure supplement 2—source data 1.** The residuals of the full model (LMA, Waking and LMA*Waking) explaining cortical temperature.
DOI: https://doi.org/10.7554/eLife.43400.010

more of the variance in cortical temperature was explained by waking in both genotypes. This was quantified with a paired t-test on Fisher Z-transformed $R^2$-values from each individual mouse's partial correlation on hourly waking and cortical temperature, and on hourly locomotor activity and cortical temperature (WT: t(5)=5.1, p=0.004; KO: t(5)=10.7, p=0.0001; see also *Figure 3*-D for $R^2$-partial correlation coefficients which are based on hourly data from all WT mice combined). We then determined the residual variance by calculating the difference between the observed cortical temperature and the temperature predicted based on the time-spent-awake in a given hour. During the light and dark phases, linear regression over- and underestimated cortical temperature, respectively (*Figure 3*-E,F; baseline 1 and 2), leading to negative residuals during the light and positive residuals during the dark phases. Fitting a sinewave through these residuals across 2 baseline days revealed a 'circadian' distribution, with a mean amplitude of 0.29°C, almost twice what was previously reported in the rat [0.15°C] (*Franken et al., 1992*). Interestingly, when considering the time course of the residuals throughout the experiment including the sleep deprivation and recovery, a consistent parallel became evident in the distribution of locomotor activity expressed per unit of waking (*Figure 3*-G).

Therefore, to determine if including locomotor activity could predict a larger portion of the variance in cortical temperature, we applied three Mixed Linear Models, with locomotor activity expressed per unit of waking (LMA/Waking). Model1 explained the variance in cortical temperature based on waking alone, Model2 also incorporated LMA/Waking, and Model3 considered additionally the interaction between Waking and LMA/Waking. Indeed, Model3 provided the best prediction of cortical temperature variance, although the improvement was marginal over the two other models (Model1: $R^2_c$=0.84; Model2: $R^2_c$=0.85; Model3: $R^2_c$=0.86; chi-squared test: Model1 vs Model2: $X^2(5)$ =16.2; p<0.0001; Model2 vs Model3: $X^2(6)$=25.0; p<0.0001). Thus, the sleep-wake distribution is the most important determinant of cortical temperature while locomotor activity is modestly contributing as well. Nevertheless, the residuals of this model, depicted in *Figure 3—figure supplement 1*, still showed a similar pattern as those in *Figure 3*-F, illustrating the contribution of other (circadian) variables and/or a non-linearity of the association between locomotor activity and sleep-wake states to changes in cortical temperature.

## The influence of sleep deprivation and CIRBP on transcripts in cortex and liver

After establishing that the sleep-wake distribution was the major determinant of cortical temperature changes in mice, we assessed whether the sleep-deprivation incurred decrease in *Cirbp* expression is responsible for the changes in clock-gene expression. To achieve this, we quantified 11 transcripts from liver and 15 from cortex before and after sleep deprivation using RT-qPCR. Genes of interest included transcripts affected by sleep deprivation (*Maret et al., 2007*; *Mongrain et al.,*

*2010*) and/or the presence of CIRBP (*Liu et al., 2013*; *Morf et al., 2012*), with an emphasis on clock genes. Mice were sacrificed either before (ZT0) or after sleep deprivation (ZT6), together with non-sleep-deprived control mice that could sleep *ad lib* (ZT6-NSD). Statistics on ZT0 (t-test) and ZT6 (two-way ANOVA) can be found in *Table 1*.

From ZT0 to ZT6, cortical temperature decreased, because mice spend more time asleep compared to preceding hours spent in the dark phase (see also *Figure 3*-A). In WT mice, this decrease was accompanied by an expected increase in the expression of the cold-induced transcript *Cirbp* (cortex: t(8)=3.2, p=0.01; liver: t(8)=2.7, p=0.03; *Figure 4*-A and *Figure 4—figure supplement 1*, compare also with the time course of *Cirbp* expression in *Figure 1*). In contrast, a 6 hr sleep deprivation caused a decrease in cortical and hepatic *Cirbp* expression relative to levels in non-sleep deprived controls (cortex: *Figure 4*-A; liver: *Figure 4—figure supplement 1*). This was consistent

**Table 1.** Statistics on RT-qPCR results.

| | Cortex | | | | Liver | | | |
| | | ZT6 | | | | ZT6 | | |
| Transcript | ZT0 | SD/NSD | GT | Interaction | ZT0 | SD/NSD | GT | Interaction |
|---|---|---|---|---|---|---|---|---|
| *Cirbp* | X | t(8)=4.9; p=0.001 | X | X | X | t(8)=4.0, p=0.004 | X | X |
| *Clock* | t = 0.86; p=0.21 | F = 2.38; p=0.14 | F = 0.85; p=0.37 | F = 8.02; p=0.01 | t = 0.03; p=0.98 | F = 0.09; p=0.77 | F = 0.81; p=0.38 | F = 0.03; p=0.87 |
| *Dbp* | t = 0.13; p=0.90 | F = 82.0; p<0.0001 | F = 0.39; p=0.54 | F = 3.06; p=0.10 | t = 1.99; p=0.08 | F = 4.37; p=0.05 | F = 0.23; p=0.64 | F = 0.0002; p=0.99 |
| *Dusp4* | t = 1.29; p=0.23 | F = 97.55; p<0.0001 | F = 0.50; p=0.49 | F = 3.24; p=0.09 | X | X | X | X |
| *Homer1a* | t = 0.96; p=0.36 | F = 228.8; p<0.0001 | F = 0.005; p=0.94 | F = 1.08; p=0.31 | X | X | X | X |
| *Hsf1* | t = 0.67; p=0.52 | F = 18.22; p=0.0006 | F = 1.79; p=0.20 | F = 1.9; p=0.18 | t = 0.14; p=0.89 | F = 3.43; p=0.08 | F = 4.63; p=0.05 | F = 0.48; p=0.50 |
| *Hsp90b1* | t = 1.29; p=0.23 | F = 7.18; p=0.0164 | F = 1.40; p=0.25 | F = 6.86; p=0.02 | t = 1.71; p=0.12 | F = 0.93; p=0.35 | F = 0.80; p=0.38 | F = 0.07; p=0.80 |
| *Hspa5* | t = 0.89; p=0.40 | F = 72.03; p<0.0001 | F = 0.03; p=0.86 | F = 5.32; p=0.03 | t = 2.02; p=0.08 | F = 0.62; p=0.44 | F = 0.84; p=0.37 | F = 0.04; p=0.86 |
| *Npas2* | t = 0.86; p=0.41 | F = 1.56; p=0.2298 | F = 0.0008; p=0.98 | F = 3.99; p=0.06 | X | X | X | X |
| *Per2* | t = 2.78; p=0.02 | F = 75.22; p<0.0001 | F = 4.78; p=0.04 | F = 0.06; p=0.80 | t = 0.90; p=0.40 | F = 0.95; p=0.34 | F = 0.01; p=0.92 | F = 0.02; p=0.90 |
| *Rbm3-short* | t = 0.05 ; p=0.96 | F = 32.04 ; p<0.001 | F = 0.31; p=0.59 | F = 0.13; p=0.73 | t = 2.23; p=0.06 | F = 47.6; p<0.0001 | F = 2.7; p=0.12 | F = 1.6; p=0.22 |
| *Rbm3-long* | t = 0.10 ; p=0.92 | F = 9.49 ; p=0.007 | F = 0.03; p=0.86 | F = 0.32; p=0.58 | X | X | X | X |
| *Nr1d1* | t = 0.91; p=0.39 | F = 8.95; p=0.009 | F = 1.09; p=0.31 | F = 6.80; p=0.02 | t = 1.59; p=0.15 | F = 31.13; p<0.0001 | F = 2.41; p=0.14 | F = 1.37; p=0.26 |
| *Sfpq* | t = 1.51; p=0.17 | F = 11.61; p=0.004 | F = 0.017; p=0.90 | F = 4.44; p=0.05 | t = 0.93; p=0.38 | F = 1.26; p=0.28 | F = 2.78; p=0.11 | F < 0.001; p=0.98 |
| *Sirt1* | t = 2.56; p=0.04 | F = 1.61; p=0.22 | F = 0.14; p=0.72 | F = 2.07; p=0.17 | t = 1.75; p=0.12 | F = 0.94; p=0.35 | F = 0.03; p=0.87 | F = 0.12; p=0.73 |

GT: genotype, SD/NSD: Sleep deprived / non-sleep deprived (control).

ZT0: t-test, degrees of freedom (df): 8.

ZT6: two-way ANOVA (factors SD and GT), df = 1 for both factors SD, GT and its interaction; error df = 16.

X: Ct >30 or undetected; Ct = Cycle threshold of the qPCR assay.

Blue: significant decrease (at ZT0: KO relative to WT; at ZT6, SD/NSD: SD relative to NSD; GT: KO relative to WT).

Red: significant increase (at ZT0: KO relative to WT; at ZT6, SD/NSD: SD relative to NSD; GT: KO relative to WT).

Purple: significant interaction. Significance level: α = 0.05.

DOI: https://doi.org/10.7554/eLife.43400.012

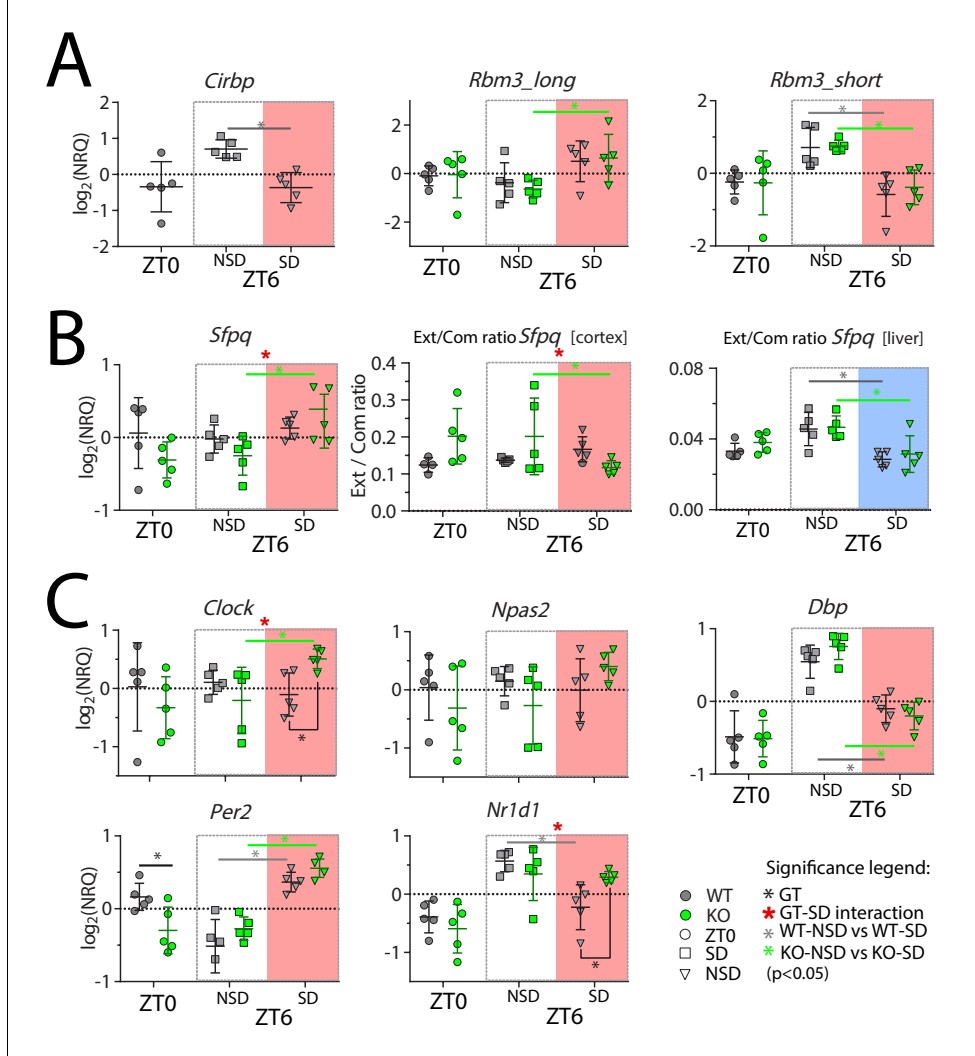

**Figure 4.** Cortical expression of several genes is affected by SD and by the lack of CIRBP. NRQ: Normalized Relative Quantity, SD: sleep deprivation (salmon and blue areas for cortex and liver, respectively), NSD: non-sleep deprived (controls), GT: Genotype, ZT: Zeitgeber Time. n = 5 for each group, each symbol represents an observation in one mouse. Mice were sacrificed at ZT0, at ZT6 after sleep deprivation (ZT6-SD) or after sleeping *ad lib* (ZT6-NSD). Statistics are performed separately for ZT0 (factor GT, t-test), and ZT6 (factor GT and SD; two-way ANOVA). Significant (p<0.05) GT differences are indicated by a black line and *, the effect of SD in WT mice with a grey line and *, and in KO mice with a green line and *. GT-SD interactions at ZT6 are indicated by a red *. See *Table 1* for statistics.

DOI: https://doi.org/10.7554/eLife.43400.013

The following source data and figure supplements are available for figure 4:

**Source data 1.** Cortical expression of transcripts in Cirbp WT and KO mice.
DOI: https://doi.org/10.7554/eLife.43400.018
**Figure supplement 1.** Changes in transcripts incurred by the absence of CIRBP and/or sleep deprivation in the liver.
DOI: https://doi.org/10.7554/eLife.43400.014
**Figure supplement 1—source data 1.** Hepatic expression of transcripts in Cirbp WT and KO mice.
DOI: https://doi.org/10.7554/eLife.43400.015
**Figure supplement 2.** Changes in transcripts incurred by the absence of CIRBP and/or sleep deprivation in the cortex.
DOI: https://doi.org/10.7554/eLife.43400.016
**Figure supplement 2—source data 1.** Cortical expression of transcripts in Cirbp WT and KO mice.
DOI: https://doi.org/10.7554/eLife.43400.017

with the wake-induced increase in cortical temperature during sleep deprivation. No *Cirbp* mRNA was detected in KO mice.

RBM3 (RNA binding motif protein 3) is another cold-inducible RNA Binding Protein that, like CIRBP, transfers circadian cycles of temperature into high-amplitude clock gene expression *in vitro* (*Liu et al., 2013*). A long and a short isoform of *Rbm3* (*Rbm3-long* and *–short*, resp.), that differ in their 3'UTR length, were discovered in the mouse cortex. Although both isoforms are referred to as 'cold-induced', they exhibit opposite responses to sleep deprivation (*Wang et al., 2010*), with a decrease in the short and an increase in the long isoform. We found that overall, the short isoform was more common than the long isoform in the cortex (PCR cycle detection number for all samples pooled: cortex: *Rbm3-short*: 25.6 ± 0.2, *Rbm3-long*: 29.7 ± 0.1, amplification efficiency *Rbm3-short*: 2.11 and *Rbm3-long:* 2.07). In the liver, only the short isoform was detected (liver: *Rbm3-short*: 28.2 ± 0.2, *Rbm3-long*:>32; that is beyond reliable detection limit). In addition, we confirmed that after sleep deprivation *Rbm3-short* expression decreased in the cortex (*Figure 4*-A) and liver (*Figure 4—figure supplement 1*), whereas *Rbm3-long* increased in cortex. The latter observation reached significance only in the KO mice (*Figure 4*-A).

As anticipated, cortical expression of the waking-induced transcripts *Homer1a*, *Dusp4, Hspa5*, *Hsp90b1* and *Hsf1* was increased by sleep deprivation (*Figure 4—figure supplement 2*). Post-hoc tests revealed that the latter two were significantly increased only in *Cirbp* KO mice. Furthermore, the effect of sleep deprivation on the transcripts *Hsp90b1* and *Hspa5* was significantly amplified in *Cirbp* KO mice compared to WT mice. Unexpectedly, no changes in the expression of heat shock transcripts incurred by sleep deprivation or genotype were detected in the liver (*Figure 4—figure supplement 1*).

*In vitro* studies have shown that the presence of CIRBP is associated with longer 3'UTRs of its target genes, such as the transcript splice-factor proline Q (*Sfpq*), resulting in a higher prevalence of long isoforms (extended or *ext*) over all isoforms (common or *com*), and thus an increased *ext/com* ratio (see FigS4-S5 in *Liu et al., 2013*). We therefore expected a lower *ext/com* ratio in mice lacking CIRBP. However, under baseline conditions [ZT0 and ZT6-NSD], *Cirbp* KO mice did not differ in their *ext/com* ratio from WT littermates (ZT0: liver: (t(8)=1.55, p=0.16; cortex: t(7)=2.0, p=0.09; ZT6-NSD: liver: t(8)=0.19, p=0.85, cortex: t(8)=1.4, p=0.20). Because RBM3 also determines the *ext/com* ratio (*Liu et al., 2013*), the lack of an effect of CIRBP on the *ext/com* ratio could be due to compensation by RBM3. We tested this by assessing the effect of sleep deprivation on the *ext/com* ratio as it acutely suppresses both RBM3 and CIRBP. Indeed, sleep deprivation significantly decreased the *ext/com* ratio in the liver in both genotypes (*Figure 4*-B; two-way ANOVA, factor sleep deprivation: F(1,16)=20.4, p=0.003). In the cortex of WT mice, however, we observed an unexpected non-significant increase in the *ext/com* ratio, leading to a significant genotype x sleep deprivation interaction (cortex: F(1,16)=5.25, p=0.036). Therefore, these data are inconclusive in confirming a role for CIRBP, and possibly RBM3, in the *in vivo* determination of *Sfpq*'s *ext/com* ratio.

Our main question concerned the contribution of CIRBP to sleep-wake-induced changes in clock-gene expression. Previous studies evaluating the effects of sleep deprivation on cortical clock transcripts showed a consistent increase in *Per2* and a decrease in *Dbp* and *Nr1d1*, whereas the response of *Clock* and *Npas2* varied among studies, but if any, tended to increase after sleep deprivation (reviewed in *Mang and Franken, 2015*). Indeed, in the cortex of WT mice, sleep deprivation increased cortical *Per2*, decreased *Dbp* and *Nr1d1* and did not significantly affect *Clock* and *Npas2* (*Figure 4*-C). Conform our hypothesis, CIRBP attenuated the sleep-deprivation-induced changes of cortical *Nr1d1*, a transcriptional repressor recently implicated in the sleep homeostat (*Mang et al., 2016*). This observation contrasts with the genotype-dependent changes in *Per2*, because when considering the lower levels of cortical *Per2* in *Cirbp* KO mice at ZT0, the effect of sleep deprivation was amplified in KO mice (*Figure 4*-C, two-way ANOVA, ZT0-ZT6[SD], interaction effect genotype x sleep deprivation: F(1,16)=12.4, p=0.003). Additionally, the expression of *Clock* in the cortex was significantly increased by sleep deprivation in *Cirbp* KO mice and not in WT littermates.

Compared to the cortex, the clock-gene expression in the liver appeared more resilient to the effects of sleep deprivation, as only *Dbp* and *Nr1d1* were significantly affected and not *Per2* (*Figure 4—figure supplement 1*). The lack of CIRBP did not interfere with this response or contribute to genotype-dependent changes of other (clock) gene transcripts in the liver.

Taken together, the absence of CIRBP modulated the sleep-deprivation-induced changes in the cortical expression of the clock genes *Nr1d1*, *Clock* and *Per2*. Furthermore, the expression of

transcripts in the heat shock pathway were also affected in a genotype-dependent manner by sleep deprivation.

## CIRBP contributes to sleep homeostasis

In three out of the five quantified cortical clock-gene transcripts, *Cirbp* KO mice showed a modulated response to sleep deprivation, suggesting that they could differ in their sleep homeostatic response (*Franken, 2013*). Thus, we hypothesized that *Cirbp* KO mice have differences in their sleep homeostatic process (Process S) and quantified EEG delta power [0.75–4.0 Hz] as a proxy of NREM-sleep pressure (*Daan et al., 1984*). In addition, we calculated the amount of NREM and REM sleep recovered after sleep deprivation relative to baseline sleep.

### Baseline characteristics of sleep-wake behavior do not differ between *Cirbp* KO and WT mice

During the 2 baseline days, no significant differences were observed in waking, NREM or REM sleep, both in terms of time spent during light and dark phases (*Table 2*), and in their distribution across the day (see *Figure 5*-A and *Figure 6*-A). Of note, under constant darkness we did not detect any change in circadian period length (period [hours]: WT (n = 5): 23.8 ± 0.03 and KO (n = 7): 23.8 ± 0.01).

### Sleep homeostatic processes under baseline and recovery

The overall time course of delta power was similar in both genotypes. In the dark phase, when mice spent most of their time awake and sleep pressure accumulates, NREM sleep delta power was highest. This contrasted with the end of the light phase [ZT8-12], where NREM sleep delta power reached its lowest levels of the day due to the high and sustained prevalence of NREM sleep in the preceding hours. However, delta power levels in *Cirbp* KO mice were higher when compared to their WT controls during both the baseline and recovery dark phases, reaching significance in the latter (*Figure 5*-A, second graph from top).

Differences in delta power can be attributed to changes in the dynamics of the underlying homeostatic process, Process S, and/or to changes in sleep-wake distribution. Our results support the latter possibility because *Cirbp* KO mice tended to spend less time in NREM sleep (and more time awake) during the early dark phase when compared to WT mice, reaching significance during the recovery (*Figure 5*-A; third graph from top). To test if the changes in the sleep-wake distribution could indeed explain the genotype differences in NREM sleep delta power, we estimated the increase ($\tau_i$) and decrease ($\tau_d$) rates using a simulation of Process S based on sleep-wake distribution. We assumed Process S to increase exponentially during waking and REM sleep with a time constant $\tau_i$ and to decrease during NREM sleep with a time constant $\tau_d$ (see Materials and methods, and *Franken et al., 2001* for more details). This simulation was not only able to reliably capture the overall dynamics (mean square of the measured-predicted differences, mean ±SEM: WT: 10.1 ± 0.3, KO: 10.4 ± 0.4), but also the genotype-specific delta power differences (*Figure 5*-A; top graph), while

**Table 2.** Baseline time spent in sleep-wake states, including theta-dominated waking (TDW; min), and locomotor activity (LMA; movements) per 12 hr per genotype (mean ±1 SEM), averages of BL1-2.
Two-way ANOVA (Factor GT and Light/Dark) on those same 12 hr values. Degrees of freedom for both GT and Light/Dark: df = 1; error term: df = 35.

| | WT | | KO | | Statistics (Two-way ANOVA) |
|---|---|---|---|---|---|
| | Light | Dark | Light | Dark | Factor GT x Light/Dark, df : 1,35 |
| NREM sleep | 389 ± 4 | 189 ± 10 | 376 ± 4 | 170 ± 13 | F = 0.02, p=0.89 |
| REM sleep | 70 ± 2 | 19 ± 2 | 66 ± 2 | 20 ± 2 | F = 0.83, p=0.37 |
| Total waking | 260 ± 4 | 512 ± 11 | 277 ± 5 | 530 ± 14 | F = 0.02, p=0.90 |
| TDW | 45 ± 3 | 179 ± 12 | 55 ± 5 | 192 ± 15 | F = 0.13, p=0.72 |
| LMA | 119 ± 16 | 817 ± 70 | 181 ± 26 | 1370 ± 142 | F = 7.1, p=0.01 |

DOI: https://doi.org/10.7554/eLife.43400.019

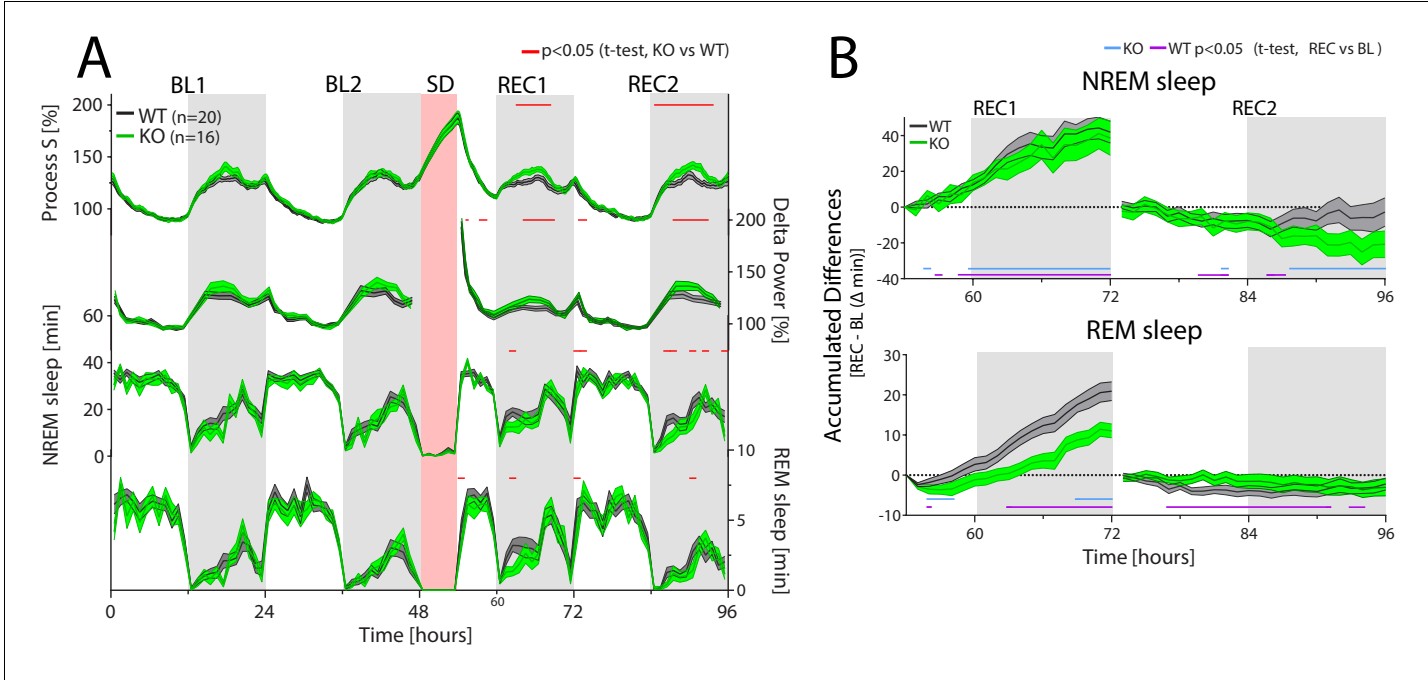

**Figure 5.** CIRBP modulates the sleep-wake distribution and REM sleep recovery after sleep deprivation. *Cirbp* KO (green lines and areas) and WT (black line, grey areas) mice during the two baseline days (BL1 and −2), sleep deprivation (SD), and the 2 recovery days (REC1 and −2; areas span ± 1 SEM range). (A) From top to bottom: Simulated delta power (Process S), measured NREM sleep delta power, NREM and REM sleep. Only during REC, both Process S and delta power are increased in *Cirbp* KO mice compared to WT (F(1,34)=5.56, p=0.024; F(1,34)=4.65, p=0.038, respectively), based on differences in NREM-sleep distribution (Genotype (GT): F(1,34)=6.02, p=0.0194). GT effects in REM sleep were also detected during recovery (factor GT: F(1,34)=5.45, p=0.026). Exact timing of GT differences is indicated by red lines above each graph (post-hoc t-test, p<0.05). (B) Top: KO mice recover as much NREM sleep as WT mice do in the first 18 hr after SD (REC1 at 72 hr: WT: 41.9 ± 6.1 KO: 38.6 ± 9.7 min; t-test: t(34)=0.30, p=0.76). Bottom: KO mice accumulated less REM sleep during the first recovery day over the baseline day in comparison to WT mice relative to baseline (REC1 at 72 hr, WT: 20.9 ± 2.3 KO: 9.9 ± 2.0, t-test: t(34)=3.7, p=0.0007). Recovery-to-baseline differences are indicated by blue (KO) and purple (WT) lines below each graph (post-hoc t-test, p<0.05). Light-grey areas mark the dark periods.

DOI: https://doi.org/10.7554/eLife.43400.020

The following source data is available for figure 5:

**Source data 1.** Simulated Process S, delta power, NREM and REM sleep in Cirbp WT and KO mice during two baseline days, a 6hr sleep deprivation and two recovery days.
DOI: https://doi.org/10.7554/eLife.43400.021

**Table 3.** Time constants, asymptotes and $S_o$ for Process S do not differ between *Cirbp* WT and KO mice.

Mean time constants (±SEM) obtained by the simulation (Process S) with the best fit to the NREM sleep delta power values, where the increase of Process S is simulated with the time constant $\tau_i$, the decrease with $\tau_d$ and the upper- and lower asymptotes by UA and LA, respectively. $S_0$ is the level of Process S at time = 0. No significant genotype differences were observed. See Material and methods for detailed description of the simulation. Degrees of freedom: df.

|  | WT | KO | t-test, df = 34 |
|---|---|---|---|
| $S_0$ [%] | 128.2 ± 2.4 | 132.1 ± 2.6 | t = 1.10, p=0.29 |
| $\tau_i$ [h] | 13.2 ± 1.2 | 12.9 ± 1.0 | t = −0.16, p=0.87 |
| $\tau_d$ [h] | 3.0 ± 0.2 | 2.8 ± 0.2 | t = −0.74, p=0.46 |
| LA [%] | 45.1 ± 1.4 | 45.1 ± 1.1 | t = −0.02, p=0.98 |
| UA [%] | 288.8 ± 3.0 | 296.6 ± 3.2 | t = 1.80, p=0.09 |

DOI: https://doi.org/10.7554/eLife.43400.028

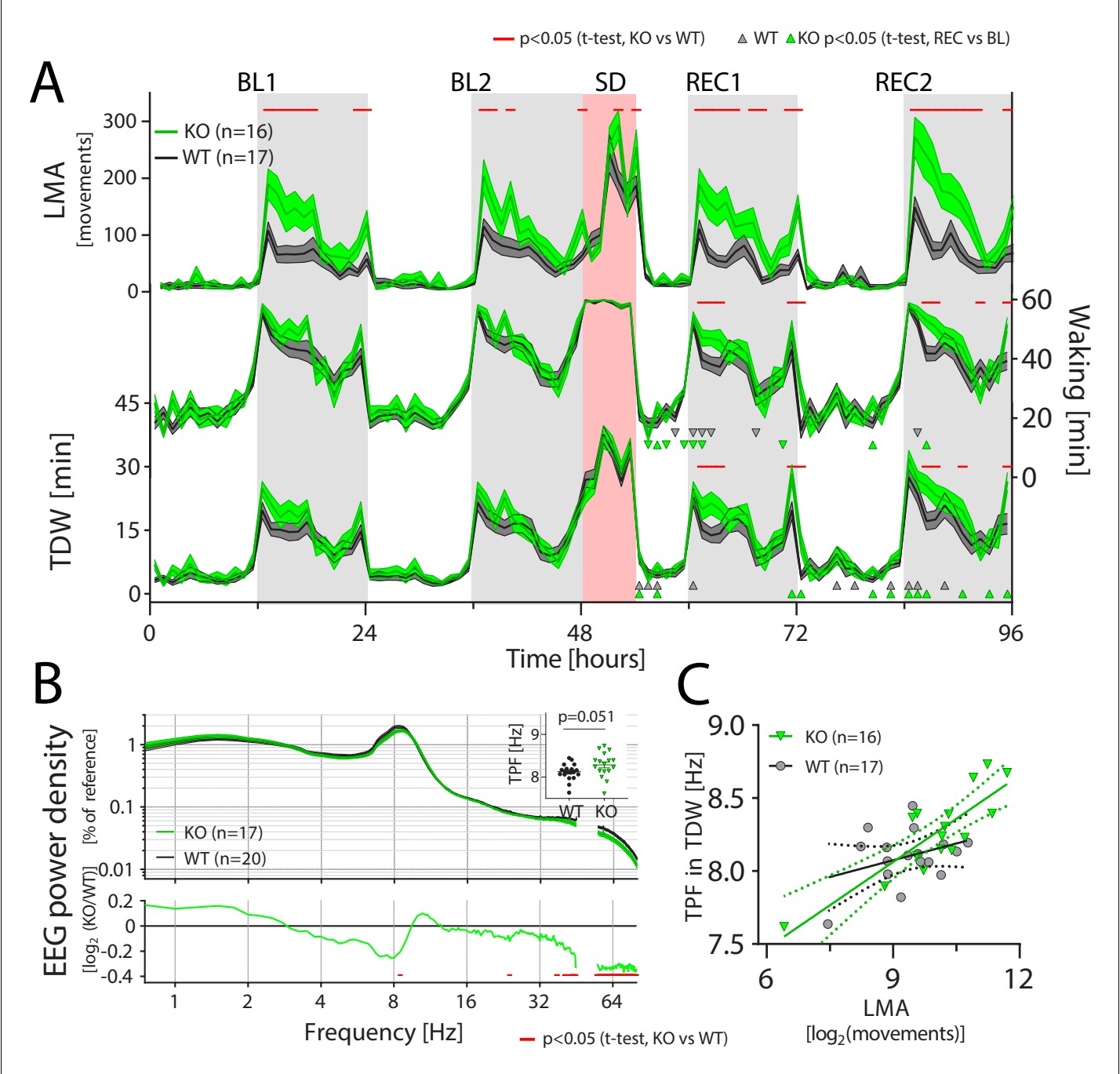

**Figure 6.** CIRBP suppresses locomotor activity and affects spectral composition during theta-dominated waking. LMA: locomotor activity, TDW: theta-dominated waking, TPF: theta-peak frequency, GT: genotype. (**A**) *Cirbp* KO (green lines and areas) and WT (black line, dark-grey areas) mice during the two baseline days (BL1 and −2), sleep deprivation (SD), and the two recovery days (REC1 and −2; areas span ± 1 SEM range). *Cirbp* KO mice are more active in the dark periods (light-grey areas) only (BL: GTxTime: F (47,1457)=3.5, p<0.0001; REC: GTxTime: F (41,1271)=5.2, p<0.001), and spent more time awake and inTDW during REC compared to WT mice (total waking: BL: GTxTime: F(47,1457) = 1.1, p=0.33 REC: GTxTime: F (41 1271)=1.9, p=0.0005; TDW: BL: GTxTime: F(47,1457)=1.1, p=0.35; REC: GTxTime: F(41,1271)=1.8, p=0.0025). Significant genotype differences are marked by red lines above each graph (post-hoc t-tests; p<0.05). Δ and ∇ indicate a significant increase and decrease in REC compared to same time in BL, respectively. (**B**) CIRBP contributes to the spectral composition of TDW in the dark phase (two-way RM ANOVA; GTxFreq: F(278,9730) = 2.0; p<0.0001, red symbols in lower panel: post-hoc t-tests, p<0.05), and KO mice tend to have faster TPF during TDW in the dark phase (t(35)=2.0; p=0.0506). (**C**) TPF in the dark phase correlates only in the KO mice significantly with LMA (WT: $R^2$ = 0.12, p=0.17, KO: $R^2$ = 0.71, p<0.0001).

DOI: https://doi.org/10.7554/eLife.43400.022

The following source data and figure supplements are available for figure 6:

*Figure 6 continued on next page*

*Figure 6 continued*

**Source data 1.** Time course of LMA, waking and theta-dominated waking in Cirbp WT and KO mice; spectral composition of theta-dominated waking, and relation between theta-peak frequency in theta-dominated waking and LMA.
DOI: https://doi.org/10.7554/eLife.43400.027
**Figure supplement 1.** Changes in the EEG spectra are observed in theta-dominated waking (TDW), but not in quiet waking.
DOI: https://doi.org/10.7554/eLife.43400.023
**Figure supplement 1—source data 1.** Spectral composition of the waking EEG in Cirbp WT and KO mice.
DOI: https://doi.org/10.7554/eLife.43400.024
**Figure supplement 2.** Slow and fast gamma power over the course of the experiment in theta-dominated waking.
DOI: https://doi.org/10.7554/eLife.43400.025
**Figure supplement 2—source data 1.** Time course of fast and slow gamma power during theta-dominated waking in Cirbp WT and KO mice.
DOI: https://doi.org/10.7554/eLife.43400.026

yielding similar Process S time constants (see *Table 3*). Thus, the reduction in NREM sleep in *Cirbp* KO mice in the beginning of the dark period caused the higher delta power values in subsequent hours, underscoring the notion that small differences in NREM sleep time can have large repercussions on delta power during periods when waking dominates and sleep pressure is high as a result (*Franken et al., 2001*).

In addition to delta power, the time spent in NREM sleep is considered another aspect of the homeostatic process, which can be quantified as the accumulated relative differences between baseline and recovery. At the end of the first recovery day, both KO and WT mice had gained ca. 40 min of NREM sleep relative to baseline (*Figure 5*-B, upper panel).

Like NREM sleep, the amount of REM sleep is also homeostatically defended (*Franken, 2002*). At the end of recovery day 1, both WT and KO mice spent more time in REM sleep compared to corresponding baseline hours. However, this increase was significantly attenuated by 46% in *Cirbp* KO mice (*Figure 5*-B, lower panel). No significant differences were detected during baseline in time spent in REM sleep (see also *Table 1*), suggesting that this attenuated rebound resulted from less REM sleep during recovery, specifically in the first hours of the dark phase when the genotypic differences were most prominent (*Figure 5*-A, lowest graph).

Thus, although CIRBP did not affect the processes underlying NREM sleep intensity and NREM sleep time, it did contribute to REM sleep homeostasis by increasing the amount of REM sleep after sleep deprivation.

## An unanticipated waking phenotype in *Cirbp* KO mice

We observed that *Cirbp* KO mice were more active than their WT littermates during the dark phase (t(31)=-2.56, p=0.015, see also *Table 2*). Specifically, *Cirbp* KO mice were almost twice as active in the 6 hr after dark onset (ZT12-18; movements: WT: 463.8 ± 60.7, KO: 801.8 ± 118.4, t(35)=-2.7, p=0.012; *Figure 6*-A). Interestingly, this increase was not associated with a significant increase in time spent awake during baseline (per 12 hr: t(35)=1.2, p=0.24, and see *Table 2*), and indeed *Cirbp* KO mice were more active per unit of waking (average in the dark phase, locomotor activity [movements/waking(min)], WT: 1.3 ± 0.13, KO: 2.1 ± 0.28; t(35)=-2.7, p=0.01). Note that cortical temperature was also not significantly increased in *Cirbp* KO mice during the dark phase (cortical temperature: WT: 35.9 ± 0.1, KO: 36.1 ± 0.1, t-test, t(10)=1.3, p=0.24) despite increased locomotor activity at this time of the day, again underscoring its minimal contribution to cortical temperature.

Because *Cirbp* KO mice were not more awake (*Table 2* and *Figure 6*), we wondered if their increased locomotor activity was associated with differences in the prevalence of waking sub-states. Theta-dominated waking is correlated with activity, most prevalent during the dark phase and sleep deprivation, and characterized by the presence of EEG theta-activity (*Buzsáki, 2006*; *Vassalli and Franken, 2017*). Despite their increased locomotor activity, *Cirbp* KO mice did not spend more time in theta-dominated waking during the dark phase of baseline (see *Table 2*, t(31)=-1.22, p=0.23). Did the increased locomotor activity in *Cirbp* KO mice relate to changes in brain activity during dark phase theta-dominated waking?

Although theta-dominated waking EEG in both genotypes showed its characteristic theta activity [6.5–12.0 Hz], subtle differences between genotypes were detected in the spectral composition of the EEG signal. In *Cirbp* KO mice, slow [32–45 Hz] and fast [55–80 Hz] gamma power were both

reduced during theta-dominated waking (*Figure 6*-B), and this reduction was observed throughout the experiment (*Figure 6—figure supplement 1*; and see time course in *Figure 6—figure supplement 2*), indicating that these spectral differences are robust across different lighting conditions, circadian time and throughout the sleep deprivation. In contrast, EEG spectral composition during 'quiet' waking (i.e. not theta-dominated waking), was remarkably similar between the two genotypes (*Figure 6—figure supplement 1*). This similarity demonstrated that changes in theta-dominated waking EEG spectra were not the result of a general CIRBP effect on the waking EEG, but specifically affected this waking sub-state.

We also observed a significant decrease in slow and a non-significant increase in fast theta activity in the theta-dominated waking EEG of *Cirbp* KO mice, which hinted at an acceleration of the theta oscillation (lower panel in *Figure 6*-B). Indeed, theta-peak frequency in this sub-state in baseline was increased in KO mice (+0.15 Hz), although the significance threshold was not met (t(35)=2.0, p=0.0506). During REM sleep, the other sleep-wake state characterized by distinct theta oscillations, theta-peak frequency was not affected (WT: 7.43 ± 0.06; KO: 7.56 ± 0.05, t(35)=1.7, p=0.10). Increased locomotor activity correlates with increased theta-peak frequency (*Jeewajee et al., 2008*). In accordance with this observation, mean $\log_2$-transformed locomotor activity in each mouse during the dark phase predicted the mean theta-peak frequency during theta-dominated waking at the same time of day (WT and KO combined; $R^2$ = 0.52, p<0.0001), although this relationship remained significant only in KO mice when assessing the two genotypes separately (*Figure 6*-C). However, this genotype-dependent association between theta-peak frequency and locomotor activity was not confirmed by a significant difference in slope between the genotypes (ANCOVA, F(1,29)=3.8, p=0.059).

Because the group correlation did not account for inter-individual differences in locomotor activity levels, we also assessed the correlations between this variable and theta-peak frequency within individual mice. To test if this association depended on lighting condition, we analyzed the dark and light phases separately (i.e. 24 values per mouse per lighting condition; see also *Table 4*). In the dark phase, this correlation was significant in all but one mouse (a KO), and both the slope and the predictive power of this correlation did not significantly differ between genotypes (slope: WT: 0.15 ± 0.01, KO: 0.14 ± 0.01, t(31)=0.37, p=0.72; $R^2$: WT: 0.81 ± 0.03; KO: 0.79 ± 0.04, t-test on the Fisher Z-transformed $R^2$-values: t(31)=-0.49, p=0.62). In the light phase, this association was weaker (dark vs. light: paired t-test: slope: t(32)=7.8, p<0.0001; Fisher Z-transformed $R^2$-values: t(32)=5.9, p<0.0001), but again did not differ between genotypes (WT: 0.07 ± 0.01, KO: 0.09 ± 0.01, t(31)=1.2, p=0.23; $R^2$: WT: 0.59 ± 0.04; KO: 0.67 ± 0.05, t-test on the Fisher Z-transformed $R^2$-values: t(31)=1.2, p=0.23). However, during the light phases, when locomotor activity and theta-dominated waking were substantially reduced and estimates of theta-peak frequency less precise, we found more non-significant associations in both genotypes (KO: 3/16; WT: 3/17 mice). Taken together, these results provide further evidence that CIRBP, through its effects on locomotor activity, reduces theta-peak frequency during wakefulness.

After establishing these genotype differences under baseline conditions, we assessed the effect of genotype on the same wake-related variables during recovery from sleep deprivation. Sleep deprivation altered the waking distribution during recovery relative to baseline (three-way RM ANOVA, factors time, genotype and baseline/recovery, factor baseline/recovery: recovery 1: F(1,558)=42.7, p<0.0001; recovery 2: F(1,1514) = 441.8, p<0.0001; see triangles in recovery 1 and 2, *Figure 6*-A). Surprisingly, while time spent awake was decreased compared to baseline, we observed several intervals during the recovery in which theta-dominated waking was increased for both genotypes (three-way RM ANOVA, factors time, genotype and baseline/recovery, factor baseline/recovery: recovery 1: F(1,558)=13.9, p=0.0002; recovery 2: F(1,1514) = 233.8, p<0.0001; *Figure 6*-A, upwards pointing triangles). Moreover, genotype differences in total waking and theta-dominated waking,

**Table 4.** Theta-peak frequency (TPF; mean ±SEM in [Hz]) during theta-dominated waking (TDW) in the baseline light and dark periods.

| TPF during BL | WT | KO |
|---|---|---|
| Light | 7.77 ± 0.03 | 7.64 ± 0.04 |
| Dark | 8.13 ± 0.04 | 8.28 ± 0.07 |

DOI: https://doi.org/10.7554/eLife.43400.029

became significant during the dark phases of both recovery days, with *Cirbp* KO mice spending more time in both than WT mice (*Figure 6*-A; see post-hoc tests indicated by red line), as if sleep deprivation amplified these non-significant genotype differences during baseline (three-way RM ANOVA on hourly values: factor genotype x time x sleep deprivation: total waking: $F_{(41,1271)} = 1.4$, p=0.04; theta-dominated waking: $F_{(41,1271)} = 1.4$, p=0.056; but not for locomotor activity $F_{(41,1271)} = 1.0$, p=0.48).

The EEG spectra during theta-dominated waking in recovery days 1 and 2 showed profiles similar to baseline (see *Figure 6—figure supplement 1*), although suggestive changes under baseline conditions reached significance during recovery, such as the increase in delta power. Similarly, the non-significant increase in theta-peak frequency in *Cirbp* KO mice during the baseline dark phases became significant during recovery (recovery 1: WT: 8.1 ± 0.05, KO: 8.4 ± 0.07, t(35)=2.7, p=0.01; recovery 2: WT: 8.2 ± 0.05, KO: 8.5 ± 0.08, t(35)=2.6, p=0.01). Furthermore, the non-significant genotype difference in slope between theta-peak frequency and locomotor activity during baseline (*Figure 6*-C), became significant after sleep deprivation (ANCOVA, $F_{(1,29)}=5.8$, p=0.02), providing further evidence that baseline genotype differences became more pronounced after challenging the sleep homeostat.

Taken together, *Cirbp* KO mice were more active than WT mice during the dark phase, which contributed to the faster theta-peak frequency. Moreover, EEG gamma power during theta-dominated waking was reduced in KO mice and the 6 hr sleep deprivation strengthened genotype differences in the sleep-wake distribution and EEG activity.

## Discussion

In this study, we showed that, like in other rodents, the sleep-wake distribution drives cortical temperature changes in the mouse. Because of the well-established link between temperature and CIRBP levels, it is likely that also the sleep-wake driven changes in brain temperature drive *Cirbp* expression. As predicted, the sleep-deprivation incurred changes in the expression of clock genes was modulated by the presence of CIRBP. However, we only observed the expected sleep-deprivation attenuated response for *Nr1d1*, whereas changes in the expression of *Per2* and *Clock* were amplified compared to WT mice. Moreover, we found evidence of altered REM sleep homeostasis in *Cirbp* KO mice. Unexpectedly, *Cirbp* KO mice were more active during the dark phase, and EEG during theta-dominated waking was characterized by a reduction of gamma activity and faster theta oscillations.

### Changes in cortical temperature are sleep-wake driven

When sleep and waking occur at their appropriate circadian times, the changes in both brain and body temperature have a clear 24-hr rhythm and therefore appear as though they are being controlled directly by the circadian clock. However, sleep-wake cycles contribute significantly to those daily changes in temperature. In humans, this involvement is powerfully illustrated by spontaneous desynchrony, where body temperature follows both a circadian and an activity-rest (and presumably, sleep-wake) dependent rhythm (*Wever, 1979*). The contribution of sleep-wake state to the daily dynamics in body temperature is further supported by forced desynchrony studies, such as (*Dijk and Czeisler, 1995*), estimating that 'masking' effects of rest-activity and sleep-wake cycles contributed between 30% and 50% to the amplitude of the circadian body temperature rhythm (*Hiddinga et al., 1997*; *Dijk et al., 2000*). Not only in humans but also in smaller mammals like rats, a circadian and rest-activity component contribute to the circadian fluctuations in body temperature (*Cambras et al., 2007*). Thus, the apparent circadian amplitude of body temperature is amplified when wake and sleep occur at the appropriate phase of the circadian rhythm.

In contrast to body temperature, brain temperature in rodents is much more determined by sleep-wake state: 80% of its variance can be explained by the sleep-wake distribution (*Franken et al., 1992* and this study). Likewise, the sleep-wake-driven changes in brain temperature are still present in arrhythmic animals (*Edgar et al., 1993*; *Baker et al., 2005*), pointing to a more important sleep-wake dependency of brain temperature compared to body temperature. Our study assessed the contribution of locomotor activity and found that waking with higher locomotor activity was associated with higher cortical temperature. Although significant, the contribution of locomotor activity to the daily changes in cortical temperature was modest and explained only 2% more of the

variance compared to waking alone. Can we optimize the prediction of cortical temperature? A non-linear relationship between sleep-wake state and cortical temperature was assumed previously (*Franken et al., 1992*) and could have improved the prediction of our model further. This is supported by the residuals from the complete model (see *Figure 3—figure supplement 2*), which exhibit under baseline conditions, a circadian distribution, whereas during sleep deprivation they remain increased as during the dark phase. Thus, the model overestimates cortical temperature during periods with little waking (light phase) and underestimates it when waking is dominant (dark phase and sleep deprivation), suggesting a non-linear relationship between these two variables.

It is important to consider that the influence of locomotor activity on cortical temperature is likely affected by the type of activity; for example, rats using a running wheel can increase their brain temperature by 2°C within 30 min (*Fuller et al., 1998*). In addition, exercise in humans leads to an increase in (proxies of) brain temperature (*Nybo et al., 2002*). Thus, although in our study the effect of locomotor activity on cortical temperature was modest compared to the effect of waking, these contributions likely differ depending on the type of physical activity.

## Locomotor activity-dependent and -independent changes in waking characteristics

Little is known about the role of CIRBP in neuronal and behavioral functioning. It was therefore unexpected that *Cirbp* KO mice were not only more active during the dark phase but also showed changes in neuronal oscillations during theta-dominated waking: a reduction in low- and high gamma power and an increase in theta-peak frequency. Because running speed correlates positively with hippocampal theta-peak frequency (*Jeewajee et al., 2008*), and our measured theta activity is mainly of hippocampal origin (*Buzsáki, 2006*), we indeed can relate this increase in speed of theta oscillations to increased locomotor activity in KO mice. In contrast to theta-peak frequency, the literature has not consistently reported on a relation between a general decrease in gamma power during active waking and its relation to locomotor activity. Some studies have found that increased speed of movement and gamma power are related (*Furth et al., 2017*; *Niell and Stryker, 2010*; *Vinck et al., 2015*), whereas others found that this association is only present in higher gamma frequencies [>60 Hz] (*Zheng et al., 2015*). Thus, it is unclear if locomotor activity relates to changes in gamma power. However, there is a clear increase in high gamma power specifically during the sleep deprivation (see *Figure 6—figure supplement 2*), as noted previously (*Vassalli and Franken, 2017*). This increase was present in both genotypes suggesting that while KO mice have a reduced capacity to produce fast gamma activity, sleep deprivation is still able to activate their fast-gamma circuitry. These results, together with the observation that during the light phase decreased gamma power was still present at a time of day when locomotor activity did not significantly differ, argues against an association between the decreased gamma activity and increased locomotor activity in *Cirbp* KO mice.

Interestingly, gamma oscillations are associated with a variety of cognitive processes (reviewed in *Bosman et al., 2014*). This is further supported by associations between behavioral impairments and changes in gamma power. For example, mice with abnormal interneurons are impaired at the behavioral level (*e.g.* lack of cognitive flexibility) and have a reduction in task-evoked gamma power in their EEG. Pharmacological stimulation of inhibitory GABA-neurons augmented power in the gamma band and rescued the behavioral phenotype of the mutants (*Cho et al., 2015*).

In the hippocampus, gamma-theta coupling, that is the occurrence of gamma oscillations at a specific phase of the theta oscillation, has been suggested to aid processes underlying memory (for review see *Colgin, 2015*). Because CIRBP reduces theta-peak frequency and increases EEG power in the gamma bands, further experiments could address if *Cirbp* KO mice have altered phase coherence between these two frequency bands. Together with the postulated function of gamma power in cognitive flexibility, it would be also be of interest to assess if the EEG phenotypes we observed in *Cirbp* KO mice are associated with cognitive abnormalities.

Several aspects of waking that appeared to differ between *Cirbp* KO and WT mice under baseline dark conditions but were non-significant, reached significance during the recovery dark phase. For example, during the baseline, *Cirbp* KO mice were 4% more awake and spent 13% more time in theta-dominated waking compared to their WT littermates. These genotype differences increased to 8% and 20%, respectively, during recovery. Similarly, theta-peak frequency and its genotype-dependent association with locomotor activity reached significance during the recovery. This suggests that

sleep deprivation amplified genotypic differences. Other sleep deprivation studies found evidence for similar phenomena, where sleep disturbance can amplify molecular and behavioral phenotypes of Alzheimer's mouse models (for review, see *Musiek and Holtzman, 2016*) and sensitivity to pain (*Sutton and Opp, 2014*). Our data indicates that sleep deprivation has a comparable effect on a number of wake-related phenotypes in *Cirbp* KO mice. It would be interesting to determine the dynamics of this change; that is if they are reversible, and if a second sleep deprivation could augment genotypic differences further.

## CIRBP adjusts clock-gene expression and REM-sleep recovery following sleep deprivation

CIRBP modulated the cortical response to sleep deprivation in the expression of three out of the five quantified clock genes. As anticipated, the sleep-deprivation incurred decrease in cortical *Nr1d1* was attenuated in *Cirbp* KO mice. NR1D1 acts as a transcriptional repressor of positive clock elements such as ARNTL (*Preitner et al., 2002*). Mice lacking both *Nr1d1* and its homolog *Rev-Erbβ* (*Nr1d2*) have a shorter and unstable period under constant conditions and deregulated lipid metabolism (*Cho et al., 2012*). We recently established that *Nr1d1* also contributes to several aspects of sleep homeostasis: *Nr1d1* KO mice accumulate NREM sleep need at a slower rate and have reduced efficiency of REM-sleep recovery in the first hours after sleep deprivation (*Mang et al., 2016*).

The cortical expression of the clock genes *Per2* and *Clock* was also modulated in the absence of CIRBP, suggesting that parts of the core clock are sensitive to its presence in response to sleep deprivation. Importantly, the effect of sleep deprivation on clock-gene expression can be modulated by CIRBP directly, or indirectly through the effects of CIRBP on transcriptional clock-gene regulators that subsequently affect the expression of downstream clock genes. For example, *Npas2* KO mice showed a reduced increase in *Per2* expression in the forebrain after sleep deprivation (*Franken et al., 2006*), while *Cry1,2* double-KO mice display a larger increase in *Per2* expression after sleep deprivation (*Wisor et al., 2008*). Thus, differences in clock-gene circuitry, as suggested by *in vitro* data (*Liu et al., 2013*; *Morf et al., 2012*), could also have contributed to the observed changes in clock-gene expression after sleep deprivation in *Cirbp* KO.

Given the role of clock genes in sleep homeostasis (*Franken, 2013*), the modulation of clock-gene expression in KO mice could have contributed to the observed REM sleep homeostatic sleep phenotype. This is supported by studies showing that mutations in clock genes incurred a loss in REM-sleep recovery (i.e. *Clock* [*Naylor et al., 2000*]), or impacted the initial efficiency of REM-sleep recovery (i.e. *Dbp* [*Franken et al., 2000*], *Per3* [*Hasan et al., 2011*], and *Nr1d1* [*Mang et al., 2016*]). Follow-up studies should address if indeed the changes in clock-gene expression in *Cirbp* KO mice are functionally implicated in this REM-sleep phenotype.

Other aspects of the homeostatic regulation of sleep, such as NREM sleep EEG delta power and time spent in NREM sleep after sleep deprivation, were unaffected in *Cirbp* KO mice. Thus, CIRBP participates specifically in REM-sleep homeostasis, whereas we do not find evidence for its contribution to NREM-sleep homeostatic mechanisms.

## Other mechanisms linking sleep-wake state to clock-gene expression

Our results show that other pathways besides CIRBP must contribute to the sleep-wake-driven changes in clock-gene expression. Some suggestions for such pathways, apart from the clock-gene circuitry itself discussed above, are shortly discussed below, as well as considerations that could potentially account for the absence of a more widespread CIRBP-dependent change in clock-gene expression that we expected based on the previously published *in vitro* studies.

*Rbm3*, another cold-inducible transcript which is closely related to CIRBP, translates temperature information into high amplitude clock-gene expression *in vitro* (*Liu et al., 2013*). Like *Cirbp*, its cortical expression is sleep-wake driven (*Wang et al., 2010*). Thus, RBM3 is another possible mechanism through which changes in sleep-wake state are linked to changes in clock-gene expression. RBM3 might have compensated for the lack of CIRBP thereby limiting the extent by which sleep-deprivation affected clock genes in *Cirbp* KO mice. A follow-up study could address this possibility by quantifying sleep deprivation-evoked changes in clock-gene expression in *Cirbp-Rbm3* double KO mice.

Heat shock factor 1 (*Hsf1*) is a member of the heat shock pathway and *in vitro* studies have shown that it transfers temperature information to the circadian clock by initiating *Per2* transcription

through binding to *Per2*'s upstream heat shock elements (*Tamaru et al., 2011*). Under undisturbed conditions, both *Hsf1* mRNA and protein levels are constitutively expressed, but the protein exhibits daily re-localization during the dark phase to the nucleus where it acts as a transcription factor (*Reinke et al., 2008*). Interestingly, CIRBP binds to the 3'UTR of *Hsf1* transcript (*Morf et al., 2012*, see supplementary data therein), although it is unclear if this affects the transcriptional activity of HSF1. We found that sleep deprivation induced a significant increase in *Hsf1* in KO mice only, which is congruent with the observation that the expression of two other transcripts of the heat shock pathway, *Hsp90b1* and *Hspa5*, were significantly amplified in KO mice after sleep deprivation. Altogether, this suggests that the increased expression of *Per2* in KO mice might be linked to increased *Hsf1* expression and underscores the presence of other temperature (and thus sleep-wake) driven pathways that can ultimately affect clock-gene expression.

Beyond temperature, many other physiological changes occur during wakefulness that can subsequently affect clock-gene expression. This has been well documented for *Per2* which can act as a sensor of stress, light, and temperature (*Franken, 2013*; *Schibler et al., 2015*), which is especially relevant in sleep deprivation studies. Another example is oxygen consumption which varies with sleep-wake state (*Jung et al., 2011*). Changes in oxygen levels can modulate the expression of clock genes through HIF1α (*Adamovich et al., 2017*). Moreover, during sleep deprivation, corticosterone levels increase, which subsequently amplifies the expression of some, but not all, clock genes (*Mongrain et al., 2010*).

We could not corroborate the hepatic increase in heat shock transcripts (*Hsf1*, *Hsp90b1* and *Hspa5*) and in *Per2* after sleep deprivation as reported in other studies (*Diessler et al., 2018*; *Maret et al., 2007*), whereas we did confirm the sleep-deprivation-induced changes in *Cirbp*, *Rbm3-short*, *Dbp* and *Nr1d1*. We cannot readily explain this discrepancy between our current and our previous studies.

Finally, we would like to briefly address an obvious shortcoming. The hypothesis of this study is based on results obtained in a relatively simple *in vitro* model (i.e. immortalized fibroblasts) and applied to a far more complex *in vivo* model (i.e. cortices and livers of freely behaving mice). Unpublished observations on the circadian dynamics of the expression of CLOCK:ARNTL target genes in the liver, such as *Nr1d1* and *Dbp*, indicate an increased circadian amplitude in *Cirbp* KO mice; that is the opposite phenotype from that observed *in vitro* (mentioned in *Schibler et al., 2015*). This observation might be relevant in explaining the unanticipated increase in *Per2* and *Clock* expression after sleep deprivation observed in the cortex of *Cirbp* KO mice. Furthermore, we could not consistently reproduce the importance of CIRBP in determining the *ext/com* ratio of *Sfpq* (see also *Figure 4*-B). Thus, *in vitro* findings will not always predict *in vivo* results, which could explain the lack of a widespread CIRBP-dependent change in clock-gene expression after sleep deprivation.

## Conclusion

This hypothesis-driven study explored whether the sleep-deprivation-induced changes in clock-gene expression could be mediated through the cold-induced transcript CIRBP. After sleep deprivation, the cortical expression of *Nr1d1*, which we recently discovered to be of importance for the sleep homeostat (*Mang et al., 2016*), was attenuated in *Cirbp* KO mice, whereas the expression of two other clock genes, *Per2* and *Clock*, was amplified. Thus, the sleep-deprivation-induced changes in clock-gene expression are modulated by CIRBP, but not always in the anticipated direction.

A large body of evidence has shown that clock genes are crucial for metabolism (reviewed in *Panda, 2016*). This is supported by the observation that disturbance of clock-gene expression, through for example genetic manipulations in mice or shift work in humans, can lead to the development of metabolic disorders (*Rudic et al., 2004*) (*Karlsson et al., 2001*). Not only sleeping at the wrong time, but also sleeping too little or of poor quality can induce disturbed metabolic state both in rats (*Barf et al., 2010*) and humans (*Copinschi et al., 2014*). Because sleep loss affects clock-gene expression (*Franken, 2013*), we propose that this could represent a common pathway through which both sleep and circadian disturbances lead to metabolic pathologies. It is thus of importance to determine the pathways through which a disturbed sleep-wake distribution affects clock-gene expression. We show that temperature and CIRBP contribute to this process, and we identified the expression of *Nr1d1* as one of the genes affected by CIRBP. Genetic (*Delezie et al., 2012*) and pharmacological (*Solt et al., 2012*) studies have shown that this transcriptional repressor is important for

healthy metabolic functioning. Further experiments could address the metabolic consequences of the attenuated response in *Nr1d1* to sleep loss.

# Materials and methods

## Key resources table

| Reagent type (species) or resource | Designation | Source or reference | Identifiers | Additional information |
|---|---|---|---|---|
| Genetic reagent (*M. Musculus* (male)) | *Cirbp* KO; *Cirbp* WT | PMID: 22711815 | RRID:MGI:5432528 | Professor Jun Fujita (Kyoto University) |
| Sequence-based reagent | RT-qCPR primers | This paper. | | See *Table 5* |
| Commercial assay or kit | RNeasy Lipid Tissue Mini Kit 50 | Qiagen | Catalog no. 74804 | |
| Commercial assay or kit | RNeasy Plus Mini Kit 50 | Qiagen | Catalog no. 74134 | |
| Commercial assay or kit | Invitrogen Superscript II reverse transcriptase | Thermo Fisher | Catalog no. 18064022 | |
| Commercial assay or kit | TaqMan mastermix | Thermo Fisher | Catalog no. 4369510 | |

**Table 5.** Sequences of the forward and reverse primer and probe used for the RT-qPCR.

| GeneName | FwdPrimer | RevPrimer | Probe |
|---|---|---|---|
| *Cirbp* | AGGGTTCTCCAGAGGAGGAG | CCGGCTGGCATAGTAGTCTC | CGCTTTGAGTCCCGGAGTGGG |
| *Clock* | CGAGAAAGATGGACAAGTCTACTG | TCCAGTCCTGTCGAATCTCA | TGCGCAAACATAAAGAGACCACTGCA |
| *Dbp* | CGTGGAGGTGCTTAATGACCTTT | CATGGCCTGGAATGCTTGA | AACCTGATCCCGCTGATCTCGC |
| *Dusp4* | GTTCATGGAAGCCATCGAGT | CCGCTTCTTCATCATCAGGT | TCCCGATCAGCCACCATCTGC |
| *Eef1a2* | CCTGGCAAGCCCATGTGT | TCATGTCACGAACAGCAAAGC | TGAGAGCTTCTCTGACTACCCTCCACTTGGT |
| *Gadph* | TCCATGACAACTTTGGCATTG | CAGTCTTCTGGGTGGCAGTGA | AAGGGCTCATGACCACAGTCCATGC |
| *Homer1a* | GCATTGCCATTTCCACATAGG | ATGAACTTCCATATTTATCCACCTTACTT | ACA5ATT5AATT5AG5AATCATGA (*) |
| *Hsf1* | CAACAACATGGCTAGCTTCG | CTCGGTGTCATCTCTCTCAGG | TGAGCAGGGTGGCCTGGTCA |
| *Hsp90b1* | TGTACCCACATCTGCACCTC | TTGGGCATCATATCATGGAA | CGCCGCGTATTCATCACAGATGA |
| *Hspa5* | CACTTGGAATGACCCTTCG | GTTTGCCCACCTCCAATATC | TGGCAAGAACTTGATGTCCTGCTGC |
| *Npas2* | AGGAAAGGACGTCTGCTTCA | CCAAGCTATGCCTCGAAGTG | CCTGGCAACCCCGCAGTTCTTA |
| *Per2* | ATGCTCGCCATCCACAAGA | GCGGAATCGAATGGGAGAAT | ATCCTACAGGCCGGTGGACAGCC |
| *Rbm3-long* | TGATGCTGTCTTCAGGATGC | GGCCCAACACAAGTAAAGGA | TCAAGGATGAGGTAAGTATGCTATCCTTGAGC |
| *Rbm3-short* | GGCTATGACCGCTACTCAGG | CAGCAATTTGCAAGGACGAT | TGAGATGGGGCATGCACACA |
| *Nr1d1* | AGGGCACAAGCAACATTACC | CAGGCGTGCACTCCATAGT | AGGCCACGTCCCCACACACC |
| *Sfpq* | GCATTTGAAAGATGCAGTGAA | CAGGAAGACCATCTTCGTCA | TCGCCCAGTCATTGTGGAACCA |
| *Sfpq_Comm* | TGGATGTTAGCAGTTTATTGACC | GCACAAGGTACACTGCCATT | TGTAAATGGCCTGTTTGGGCAGG |
| *Sfpq_Ext* | TGCTTTCCTCCCACCATAAG | TTGCTCTAACGAAAGGAAATTCA | TGGGGATGTTTTGATGATGTCAGTTCA |
| *Sirt1* | TTGTGAAGCTGTTCGTGGAG | CTCATCAGCTGGGCACCTA | TTTTAATCAGGTAGTTCCTCGGTGCCC |
| *Tbp* | TTGACCTAAAGACCATTGCACTTC | TTCTCATGATGACTGCAGCAAA | TGCAAGAAATGCTGAATATAATCCCAAGCG |

(*) 5 = propynyl dC; increases the melting temperature of the probe.
DOI: https://doi.org/10.7554/eLife.43400.030

## Mice and housing conditions

*Cirbp* KO mice, kindly provided by Prof Jun Fujita (Kyoto University, Japan), were maintained on a C57BL6/J background. In these mice, *Cirbp* exons were replaced by a TK-neo gene through homologous recombination in D3 embryonic stem cells, resulting in the absence of the *Cirbp* transcript and protein (*Masuda et al., 2012*). Breeding couples or trios consisted of heterozygous male and female mice. WT littermates were used as controls. Throughout all the experiments, mice were individually housed in polycarbonate cages (31 × 18 × 18 cm) with food and water *ad libitum* and exposed to a 12 hr light/12 hr dark cycle (LD12:12). Light was delivered by overhead fluorescent tubes resulting in 70–90 lux at cage level. All experiments were approved by the Ethical Committee of the State of Vaud Veterinary Office Switzerland under license VD2743 and 3201.

## EEG/EMG and thermistor surgery

At the age of 9 to 13 weeks, 17 KO and 20 WT male mice were implanted with electroencephalogram (EEG) and electromyogram (EMG) electrodes (eight experimental cohorts). The surgery took place under deep xylazine/ketamine anesthesia supplemented with isoflurane (1%) when necessary (for details see *Mang and Franken, 2012*). Briefly, six gold-plated screws (diameter 1.1 mm) were screwed bilaterally into the skull over the frontal and parietal cortices. Two screws served as EEG electrodes and the remaining four anchored the electrode-connector assembly to the skull. As EMG electrodes, two gold wires were inserted into the neck musculature. Of all EEG/EMG implanted mice, 8 KO and 9 WT mice were implanted with a thermistor (serie P20AAA102M, General Electrics (currently Thermometrics), Northridge, CA) which was placed on top of the right cortex (2.5 mm lateral to the midline, 2.5 mm posterior to bregma). The EEG and EMG electrodes (and thermistor) were soldered to a connector and cemented to the skull. Mice recovered from surgery during 5–7 days before they were connected to the recording cables in their home cage for habituation, which was at least 6 days prior to the experiment. In total no less than 11 days were scheduled between surgery and start of experiment.

## Experimental protocol and data acquisition

EEG and EMG signals, cortical temperature and locomotor activity were recorded continuously for 96 hr under LD with the same lighting conditions as the housing conditions (see above). The recording started at light onset; that is Zeitgeber Time (ZT)0. During the first 48 hr (baseline days 1 and 2), mice were left undisturbed to establish a baseline. Starting at ZT0 of day 3, mice were sleep deprived by gentle handling for 6 hr (ZT0–6), as described in *Mang and Franken (2012)*. The remaining 18 hr of day 3 and the entire day 4 were considered as recovery (days REC1 and REC2, respectively). The analog EEG and EMG signals were amplified (2,000×), digitized at 2 kHz and subsequently down sampled to 200 Hz and stored. The EEG was subjected to a discrete Fourier transformation yielding power spectra (range: 0–100 Hz; frequency resolution: 0.25 Hz; time resolution: consecutive 4 s epochs; window function: Hamming). Thermistors were supplied with a constant measuring current ($I_{const}$ = 100 microA), and voltage (V) was measured at 10 Hz to calculate the median resistance ($R_t$) per 4 s epoch as in *equation. (1)*.

$$R_t = \frac{V}{I_{const}} \tag{1}$$

Each thermistor has an individual material constant, β. The resistance was measured at 25°C ($R_{25°C}$) and 37°C ($R_{37°C}$) by the manufacturer, and used to determine $β$ as in *equation.(2)*, with temperature values T in °Kelvin (°C + 273.15).

$$\beta = \frac{T_{25} * T_{37}}{T_{25} - T_{37}} * \ln \frac{R_{37°C}}{R_{25°C}} \tag{2}$$

Following on *equation. (2)*, the temperature ($t$) in °C can be calculated as described in *equation. (3)*.

$$t(°C) = \left[ \frac{1}{\beta} * \log \left[ \frac{R_t}{R_{25°C}} \right] + \frac{1}{T_{25°C}} \right]^{-1} - 273.15 \tag{3}$$

The EEG, EMG, and voltage across the thermistor were recorded with Hardware (EMBLA) and software (Somnologica-3) purchased from Medcare Flaga (EMBLA, ResMed, USA). Locomotor activity was detected with passive infrared sensors (Visonic Ltd, Tel Aviv, Israel) and quantified with ClockLab software (ClockLab, ActiMetrics, Wilmette, IL).

## Analysis of locomotor activity

To inspect the time course of locomotor activity corrected for time-spent-awake, raw locomotor activity was expressed per unit of time awake (minutes) in percentiles to which an equal amount of time-spent-awake contributed (as in *Figure 3*-G). The number of percentiles per recording period were chosen according to the prevalence of wakefulness, where six percentiles were used during the light phase and 12 during the dark phase, with the exception for six sections during the sleep deprivation and three sections during the remaining 6 hr of the light phase of recovery day 1. To assess genotype differences in locomotor activity (*Figure 6*), the absolute number of movements were inspected. The locomotor activity recordings of four mice (3 WT, 1 KO) were interrupted due to technical problems during the experiment, leaving data from 17 WT and 16 KO mice for analyses involving locomotor activity.

After the EEG-based sleep phenotyping, we determined circadian rhythms in locomotor activity under constant dark (DD) conditions lasting at least 2 weeks in 5 WT and 7 KO mice. Period length was determined by Chi-squared test with ClockLab software (ClockLab, ActiMetrics, Wilmette, IL).

## Determination of behavioral states

Offline, the mouse's behavior was visually classified as 'Wakefulness', 'REM sleep', or 'NREM sleep' for consecutive 4 s epochs based on the EEG and EMG signals, as previously described (*Mang and Franken, 2012*). Wakefulness was characterized by EEG activity of mixed frequency and low amplitude and variable muscle tone. NREM sleep was defined by synchronous activity in the delta frequency range (1–4 Hz), and low and stable muscle tone. REM sleep was characterized by regular theta oscillations (6–9 Hz) with low EMG activity. Waking was further differentiated into 'quiet waking' and 'theta-dominated waking' (TDW). Theta-dominated waking was determined based on the relative importance of power in the 6.5 to 12.0 Hz range to the overall power in the EEG of an artefact-free epoch scored as wakefulness, as described in *Vassalli and Franken (2017)*. We refer to waking that is not classified as theta-dominated waking as 'quiet' waking. Epochs containing EEG artefacts were marked according to the state in which they occurred and excluded from EEG spectral analysis but included in the sleep-wake state analyses. During the 4-day recording, $7.0 \pm 0.9\%$, $2.1 \pm 0.3\%$ and $2.5 \pm 0.2\%$ of the epochs were scored as an artefact in waking, NREM, and REM sleep, respectively, and this did not differ between genotypes (t-tests, $t(35)=1.77$, $p=0.09$; $t(35)=0.64$, $p=0.53$; $t(35)=0.99$, $p=0.33$, respectively).

## Analysis of cortical temperature

The raw cortical temperature data showed unexpected variation. Therefore, we inspected the inter-individual variation in daily amplitude and absolute cortical temperature levels. The latter was determined in two ways: i) by averaging cortical temperature during the last five hours of sleep deprivation, thus minimizing the sleep-wake state incurred differences in cortical temperature and ii) by averaging cortical temperature during the 12 hr baseline light phase. These measures were highly correlated ($R^2 = 0.99$; $p<0.0001$). Variation in the daily amplitude was quantified by averaging the difference between the highest and lowest hourly mean of cortical temperature of each of the 2 baseline days. No effect of genotype on absolute average cortical temperature or amplitude was detected (t-test, $t(12)=0.61$, $p=0.55$; $t(12)=-0.63$, $p=0.54$, respectively). Two mice (one of each genotype) exhibited a ca. 2-fold reduction in amplitude together with 2°C higher values during the sleep deprivation relative to the other mice (*Figure 7*, pink symbols). Therefore, we excluded these two mice from subsequent cortical temperature analysis. Three other mice (2 WT and 1 KO) showed normal amplitude but overall lower absolute values (*Figure 7*, blue symbols). We corrected for this difference by raising their cortical temperature values by the difference between the cortical temperature reached in each of these three mice during the sleep deprivation to the average cortical temperature reached over the same recording period in the remaining nine mice. Of note, most of our cortical temperature analyses focus on its relative sleep-wake-dependent changes, which are not

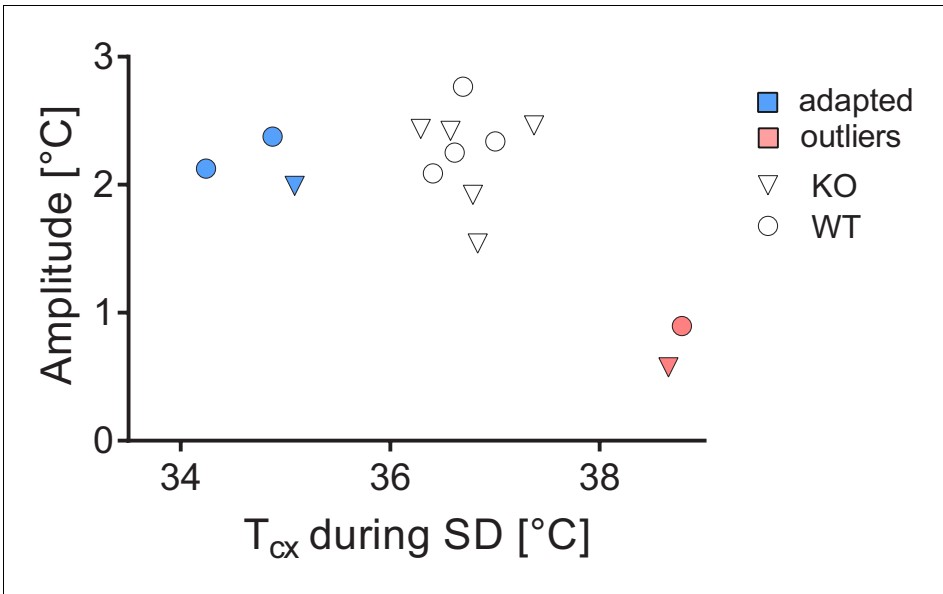

**Figure 7.** Assessment of amplitude and absolute values of cortical temperatures. Two outliers were detected (pink), whereas three others were corrected for their low values (blue).
DOI: https://doi.org/10.7554/eLife.43400.031
The following source data is available for figure 7:

**Source data 1.** The daily amplitude of cortical temperature and cortical temperature reached during sleep deprivation.
DOI: https://doi.org/10.7554/eLife.43400.032

affected by differences in absolute cortical temperature values. During the recording, one KO mouse and one WT mouse had random fluctuations of cortical temperature beyond physiological reach and were therefore excluded from analysis involving the daily dynamics of cortical temperature (*Figure 3*). In the recovery, a KO mouse was excluded due to aberrant high cortical temperature that could not be accounted for by the sleep-wake distribution, leaving 6 WT and 5 KO mice for analyses involving REC1 and REC2.

Because visualization of all 4 s epochs occurring in 24-hr day is not compatible with the resolution of *Figure 2*-A, sleep-wake states were averaged per minute and assigned to either wake, NREM or REM sleep. Because REM sleep occurs less and in shorter bouts than waking and NREM sleep, this state is slightly underrepresented in the hypnogram of *Figure 2*-A. Cortical temperature values were averaged per minute.

We analyzed cortical temperature 1.5 min before and after sleep-wake state transitions (i.e. transitioning from wake to NREM sleep, NREM to REM sleep, REM sleep to wake and NREM sleep to wake). A sleep-wake state transition was selected when the state immediately before and after the transition lasted at least eight epochs (i.e. >32 s). With this criterion, an average of 38 wake to NREM sleep, 101 NREM sleep to REM sleep, 28 REM sleep to wake and 32 NREM sleep to wake transitions per mouse during the two baseline days were detected. The profile of cortical temperature changes before and after the transition was constructed by calculating temperature relative to the mean temperature at a given sleep-wake state transition (i.e. the average of temperature in the epoch before and after the sleep-wake state transition). We subsequently constructed an individuals' average cortical temperature profile for each sleep-wake state transition. For this average, at least 10 traces were contributing at a given time point to prevent that average individual temperature profiles were based on a few transitions only. Thus, the further from the sleep-wake state transition, the less epochs contributed to the average individual cortical temperature profile. One WT mouse exhibited an extreme drop in cortical temperature (−0.2°C in a 4 s epoch) after the transition from NREM sleep to wake in its average cortical temperature trace, but not in other sleep-wake state transitions. We attributed this observation to a technical artefact and therefore this mouse was

excluded from the NREM sleep to wake and wake to NREM sleep transitions. No transitions from waking to REM sleep, nor from REM to NREM sleep are depicted because these events are relatively rare and short and were therefore not detected with our strict criteria.

The residuals of the correlation between waking and cortical temperature exhibited a circadian pattern under baseline conditions. We visualized the properties of this pattern further by fitting a sinewave through the data (Prism, non-linear regression; sine-wave with non-zero baseline; least squares fit).

## Analysis of EEG based on sleep-wake state

Unless otherwise stated, all mice (20 WT and 17 KO) were included in the analyses based on the EEG data. Spectral content of the EEG within sleep-wake states was calculated as follows. To account for inter-individual differences in overall EEG power, EEG spectra were expressed as a percentage of an individual reference value calculated as the total EEG power across 0.75–45 Hz and all sleep-wake states in the 48 hr baseline. This reference value was weighted so that for all mice the relative contribution of the three sleep-wake states (wake, NREM and REM sleep) to this reference value was equal.

Theta-peak frequency was calculated by determining the frequency at which power density peaks per 4 s epoch and subsequently averaged per individual. Power density peaks were quantified in the 6.5 to 12.0 Hz range and the 5.5 to 12.0 Hz range for theta-dominated waking and REM sleep, respectively.

Time course analysis of EEG delta power (i.e. the mean EEG power density in the 0.75–4.0 Hz range in NREM sleep) during baseline and after sleep deprivation was performed as described previously (*Franken et al., 1999*), and similar to the analysis of locomotor activity per unit of waking. The light periods of baseline days 1 and 2, and recovery day 2 were divided into 12 percentiles, the recovery day 1 light period (ZT6–12) into eight sections, and all dark periods into six sections. The timing of these percentiles was based on the prevalence of NREM sleep. EEG delta power values in NREM sleep were averaged within each percentile and then expressed relative to the mean value reached in the last 4 hr of the two main rest periods in baseline between ZT8–12. This reference was selected because delta power reaches lowest values at this time of the day and is least influenced by differences in prior history of sleep and wakefulness (see also *Franken et al., 1999*). In the time course of NREM delta power, one mouse (KO) demonstrated a strong decrease over the course of the experiment which could not be attributed to changes in the sleep-wake distribution. 9 out of the 12 delta power values during the light phase of recovery day 2 in this mouse were outliers (MAD outlier test, for details see *Leys et al., 2013*). This mouse was excluded from the analyses involving sleep homeostasis (*Figure 5*).

The effect of 6 hr sleep deprivation on subsequent time spent in NREM and REM sleep was assessed by calculating the recovery-baseline difference in sleep time per 1 hr intervals.

## Simulating NREM sleep EEG delta power [Process S]

We applied a computational method to predict the change in delta power during NREM sleep based on the sleep-wake distribution as described before (*Franken et al., 2001*). Process S is assumed to increase according to an exponential saturating function with time constant $\tau_i$ during waking and REM sleep, and exponentially decreasing with a time constant $\tau_d$ during NREM sleep (*equations. (4) and (5)*, respectively).

$$S_{t+1} = UA - (UA - S_t) * e^{-\frac{dt}{\tau_i}} \tag{4}$$

$$S_{t+1} = LA + (S_t - LA) * e^{-\frac{dt}{\tau_d}} \tag{5}$$

UA represents the upper asymptote, LA the lower asymptote and *dt* the time step of the iteration (4 s). Both asymptotes were estimated for each individual mouse. The upper asymptote was based on the 99% level of the relative frequency distribution of delta power reached in all 4 s epochs scored as NREM sleep in the 4-day recording. As an estimate of the lower asymptote, the intersection of the distribution of delta power values in NREM sleep with REM sleep was taken. At the start of the simulation, an iteration through the first 24 hr was performed with $S_0 = 150$ at t = 0. The value

reached after 24 hr is independent of $S_0$ at t = 0 and, assuming a steady state during baseline, reflects Process S at the start of the baseline for a given combination of time constants.

The fit was optimized by minimizing the mean squared difference of simulated and observed delta power for a range of $\tau_i$: 1–25 hr, step size 0.125 hr; $\tau_d$: 0.1–5.0 hr, step size 0.025 hr; that is the simulation was run for all 38'021 combinations of $\tau_i$ and $\tau_d$ for each mouse. The combination of $\tau_i$ and $\tau_d$ giving the best fit was used to assess differences in Process S between genotypes.

We noted a subtle but consistent linear discrepancy in the alignment of the simulated Process S to the measured delta power values at the end of the light phase on baseline day 1 and 2, and recovery day 2 (Pearson correlation, slope≠0: 1 sample t-test; t(35)=-4.38, p=0.0001). This change correlated well with the day-to-day changes in total spectral power in the EEG calculated across all sleep-wake states in baseline day 1 and 2, and recovery day 2 (Pearson correlation: $R^2$ = 0.70, p<0.0001; n = 36). There was no effect of genotype on the slope (Δ delta power %/h; students' t-test; t(34)=0.62; p=0.54; WT:−0.086 ± 0.027; KO:−0.065 ± 0.021) or intercept (t(34)=-0.88; p=0.38; WT: 101.5 ± 0.62; KO: 100.7 ± 0.56; WT: n = 20, KO: n = 17) of the linear correlation. We attributed these linear changes to be of non-biological origin and detrended the measured NREM sleep EEG delta power values before optimizing the fit between observed and simulated delta power.

## Gene expression in liver and brain

Five mice of each genotype (n = 15 per genotype in total) were sacrificed either prior to sleep deprivation (ZT0), at ZT6 allowing them to sleep *ad lib* (i.e. without sleep deprivation; ZT6-NSD), or at ZT6 after 6 hr sleep deprivation (ZT6-SD) across four experimental cohorts. Mice were randomly assigned to one of the three experimental conditions. Genes of interest included transcripts affected by sleep deprivation (*Maret et al., 2007*; *Mongrain et al., 2010*) and/or by the presence of CIRBP (*Liu et al., 2013*; *Morf et al., 2012*) with a special interest for clock genes. Specific forward and reverse primers and Taqman probes were designed (see *Table 5*) to quantify mRNA.

Upon sacrifice, both the cerebral cortex and liver were extracted and immediately flash frozen in liquid nitrogen. Samples were stored at −80°C. RNA from cortex was extracted and purified using the RNeasy Lipid Tissue Mini Kit 50 (QIAGEN, Hombrechtikon, Switzerland); RNA from liver was extracted and purified using the RNeasy Plus Mini Kit 50 (QIAGEN, Hombrechtikon, Switzerland), according to manufacturer's instructions. RNA quantity (NanoDrop ND-1000 spectrophotometer; Thermo Scientific, Wilmington, NC, USA) and integrity (Fragment Analyzer, Advanced Analytical, Ankeny, IA, USA) was measured and verified for each sample. 1000 ng of purified total RNA was reverse-transcribed in 20 μL using a mix of First-strand buffer, DTT 0.1M, random primers 0.25 μg/μl, dNTP 10 mM, RNAzin Plus RNase Inhibitor and Superscript II reverse transcriptase (Invitrogen, Life Technologies, Zug, Switzerland) according to manufacturers' procedures. The cDNA was diluted 10 times in Tris 10 mM pH 8.0, and 2 μL of the diluted cDNA was amplified in a 10 μL TaqMan reaction in technical triplicates on an ABI PRISM HT 7900 detection system (Applied Biosystems, Life Technologies, Zug, Switzerland). Cycler conditions were: 2 min at 50°C, 10 min at 95°C followed by 45 cycles at 95°C for 15 s and 60°C for 1 min. Standard curves were calculated to determine the amplification efficiency (E). A sample maximization strategy was used where all biological replicates of one tissue were amplified for two genes per plate. Gene expression levels were normalized to two reference genes (cortex: *Eef1a2* and *Gapdh*: M = 0.23, CV = 0.09 and liver: *Gadph* and *Tbp*; M = 0.32, CV = 0.11) using QbasePLUS software (Biogazelle, Zwijnaarde, Belgium). *Rbm3* isoforms were quantified in a separate run in liver and cortex, again with their housekeeping genes (same as previously; cortex: M = 0.22, and CV = 0.08; liver: M = 0.13, CV = 0.05). Transcripts with an average Ct-value >30 were omitted from analysis (in KO and WT livers: *Rbm3*, *Dusp4*, *Homer1a* and *Npas2*; in cortex and liver of KO mice: *Cirbp*). Results are expressed as normalized relative quantity (NRQ) which based on the overall mean expression per gene, which was set at 1.0 (*Hellemans et al., 2007*).

CIRBP affects the poly-adenylation sites of several transcripts (*Liu et al., 2013*). We explored if this newly discovered role of CIRBP could be corroborated in our study by focusing on the transcript Splicing factor, proline and glutamine rich (*Sfpq*) which exhibits CIRBP-dependent alternative poly-adenylation (APA) (*Liu et al., 2013*, see their Supplemental Figures 4-5). We calculated the ratio of the prevalence of the external 3'UTR region over the common region according to *equation. (6)*,

$$Ratio_{ext/comm} = \frac{E^{-Ct_{ext}}}{E^{-Ct_{comm}}} \tag{6}$$

where E is the amplification efficiency and $Ct_{ext}$ and $Ct_{comm}$ the number of cycles for the detection of the extended and common isoform, respectively.

## Statistics

Statistics were performed in R (version 3.3.2) and Prism (version 7.0). The threshold of significance was set at p=0.05, and all statistics were solely performed on biological replicates. To be more specific, our RT-qPCR data stems from five biological replicates, whereas amplification of cDNA from one biological replicate is performed in three technical replicates. Deviations from the mean are representing standard error of the mean. The distribution of the locomotor activity data was normalized by a $log_2$ transformation on the hourly values, allowing for subsequent parametric analyses on the relationship between cortical temperature and locomotor activity as in *Figure 3*. Time course data were analyzed by one- and two-way repeated measures (RM) analysis of variance (ANOVA) with as factors 'time' and 'genotype' (GT). Upon significance, post-hoc Fisher LSD tests were computed. Differences between baseline and recovery values within genotype were computed by paired t-tests. EEG spectra were also analyzed by one- and two-way RM ANOVA with as factors 'time' or 'frequency' and 'genotype. When genotype or its interaction with time or frequency reached significance, post-hoc t-tests were computed. The above-mentioned analyses were all performed in Prism.

Correlation coefficients of linear regression were calculated in Prism over all hourly values of locomotor activity, cortical temperature and waking per genotype (96 per mice). To compare slopes of regression lines between genotypes, an ANCOVA was applied based on (*Zar, 1984*) and run in Prism. To quantify the contribution of waking and locomotor activity independent from each other to cortical temperature, a partial correlation was performed (R software; package 'ppcor', function pcor.test). Mixed model analysis was performed with factors locomotor activity ($log_2$ transformed), waking, and genotype (R packages 'lme4', 'lmer', 'lmerTest', and' MuMIn'). Model1 quantified the predictive power of waking, Model2 of waking and locomotor activity per unit of waking (LMA/Waking) and Model3 of waking, LMA/Waking and its interaction, to predict cortical temperature. Predictive power of models was compared with Chi-squared tests by assessing the statistical significance in the reduction of residual sum of squares between two models ordered by complexity; that is Model1 was compared to Model2, and upon significance, Model2 was compared to Model3. Goodness-of-fit was assessed by the marginal R-squared ($R^2_m$) which explains the effect of the fixed factors only, and the conditional R-squared ($R^2_c$), which considers the individual variance as well and is therefore more biological relevant. Hence, in the results section only the $R^2_c$ values are reported.

For the molecular data, the qPCR NRQ values were $log_2$-transformed to normalize the distribution. Genotype differences at ZT0 were tested with a t-test. The effect of sleep deprivation and genotype at ZT6 was assessed by two-way ANOVA with post-hoc Fisher LSD tests upon significance. One outlier (WT, cortex) in the *ext/com* ratio analyses was detected by the Grubbs outliers test ($\alpha < 0.05$) and excluded.

## Acknowledgements

We thank Maxime Jan for his help in constructing the linear mixed model, our colleagues for their assistance with the sleep deprivations, Hannes Richter from the Genomic Technology Facility (University of Lausanne), for his support when setting up the RT-qPCR, David Gatfield and Bulak Arpat (University of Lausanne) for insightful discussions and Jun Fujita (Kyoto University, Japan) for sharing the *Cirbp* KO mice. This study was performed at the University of Lausanne, Switzerland, and supported by the Swiss National Science Foundation (SNF n°146694 to PF supporting MMBH) and the State of Vaud (supporting MMBH, YE, JH and PF).

## Additional information

### Funding

| Funder | Grant reference number | Author |
|---|---|---|
| Swiss National Science Foundation | 146694 | Marieke MB Hoekstra |
| Etat de Vaud | | Marieke MB Hoekstra<br>Yann Emmenegger<br>Jeffrey Hubbard<br>Paul Franken |

The funders had no role in study design, data collection and interpretation, or the decision to submit the work for publication.

### Author contributions

Marieke MB Hoekstra, Conceptualization, Data curation, Software, Formal analysis, Validation, Investigation, Visualization, Writing—original draft, Project administration, Writing—review and editing; Yann Emmenegger, Formal analysis, Investigation; Jeffrey Hubbard, Writing—review and editing, Rewrote the manuscript for both style and content following a suggestion by the reviewers, Provided critical input to the rebuttal and supported aspects of the data analyses; Paul Franken, Conceptualization, Software, Supervision, Funding acquisition, Project administration, Writing—review and editing

### Author ORCIDs

Marieke MB Hoekstra (iD) http://orcid.org/0000-0003-0723-2026
Paul Franken (iD) https://orcid.org/0000-0002-2500-2921

### Ethics

Animal experimentation: All experiments were approved by the Ethical Committee of the State of Vaud Veterinary Office Switzerland under license VD2743 and 3201.

### Decision letter and Author response

Decision letter https://doi.org/10.7554/eLife.43400.037
Author response https://doi.org/10.7554/eLife.43400.038

## Additional files

### Supplementary files

• Transparent reporting form
DOI: https://doi.org/10.7554/eLife.43400.033

### Data availability

Source data files underlying all figures have been provided.

The following previously published dataset was used:

| Author(s) | Year | Dataset title | Dataset URL | Database and Identifier |
|---|---|---|---|---|
| Maret S, Dorsaz S, Gurcel L, Pradervand S, Petit B, Pfister C, Hagenbuchle O, O'Hara BF, Franken P, Tafti M | 2007 | Molecular correlates of sleep deprivation in the brain of three inbred mouse strains in an around-the-clock experiment | https://www.ncbi.nlm.nih.gov/geo/query/acc.cgi?acc=GSE9442 | NCBI Gene Expression Omnibus, GSE9442 |

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
