## [Decision Letter]

Thank you for submitting your article "Cold Inducible RNA-binding protein (CIRBP) adjusts clock gene expression and REM sleep recovery after sleep deprivation" for consideration by *eLife*. Your article has been reviewed by three peer reviewers, including Louis J Ptáček as the Reviewing Editor and Reviewer #1, and the evaluation has been overseen by Catherine Dulac as the Senior Editor. The following individual involved in review of your submission has agreed to reveal their identity: Ying-Hui Fu (Reviewer #3).

The reviewers have discussed the reviews with one another and the Reviewing Editor has drafted this decision to help you prepare a revised submission.

Summary:

This paper by Hoekstra and colleagues studies sleep-wake regulation in *Cirbp*-deficient mice showing reduced REM sleep recovery after sleep deprivation. Data are correlated with experiments on cortical temperature regulation by sleep. The authors suggest that sleep loss-induced increases in cortical temp may down-regulate *Cirbp* and affect clock regulation in this tissue, thus modulating SD recovery responses.

The sleep and temperature analyses are very well done and the correlation between cortical temp and sleep states is striking. It is very interesting that *Cirbp* loss affects sleep deprivation responses, but not baseline sleep regulation. In addition, it appears to have an unexpected effect on locomotor activity.

The gene expression data and the functional implication of the circadian clock in this context is less well worked out. Importantly, it remains somewhat unclear by which mechanism *Cirbp* regulates clock function in the cortex and to which extent the clock machinery is mediating the effects of *Cirbp* loss in the mutant mice on sleep deprivation responses.

Essential revisions:

1) Figure 2/3: I think it is very important to show the KO data for baseline sleep regulation. How are state transients reflected in cortical temperature? Considering the postulated temp-*Cirbp*-clock-sleep axis, would one not expect differences between WTs and KOs already under these (non-SD) conditions?

2) In Figure 4, they show that in the KO mice, that there are changes in expression of several clock genes and they imply they are all regulated by CIRBP. Since CLOCK and PER are transcriptional regulators, it is possible that only one is regulated by CIRBP and that the effects on other genes are indirect (through the clock). This should be discussed.

The findings on clock gene regulation by SD are puzzling and in part contradict the results expected from the Schibler paper (Morf et al., 2012). This suggests a different mechanism of *Cirbp*-clock interaction in the cortex compared to fibroblasts. I think this is a critical point to understand the KO phenotype. Considering the differential responses of *Clock* and *Rev-Erb*, one could evaluate their contribution to SD responses by testing respective mouse mutants (and, possibly, double mutants).

3) Figure 6: Also, for locomotor activity, the specific contribution of the clock to *Cirb*-mediated effects should be tested.

In Figure 6A, it would be interesting to phenotype these mice in a skeletal photoperiod. Since they are showing differences in sleep homeostasis, it is possible that activity is being 'masked' during the light phase. What is the intensity of light in these experiments?

---

## [Author Response]

Essential revisions:1) Figure 2/3: I think it is very important to show the KO data for baseline sleep regulation. How are state transients reflected in cortical temperature? Considering the postulated temp-Cirbp-clock-sleep axis, would one not expect differences between WTs and KOs already under these (non-SD) conditions?

We thank the reviewers for raising this important issue. We have now included the cortical temperature changes across sleep-wake state transitions in KO mice in Figure 2B, and added a supplementary figure to Figure 3A showing the time course of cortical temperature in both genotypes under baseline conditions (Figure 3—figure supplement 1). Both analyses demonstrate that lack of CIRBP does not affect the strong relationship between sleep-wake state and cortical temperature. Although *Cirbp* expression is known to depend on temperature changes in a variety of tissues, we are not aware of any reports suggesting that CIRBP is implicated in the regulation of temperature. This is consistent with the lack of a genotype effect on temperature changes at sleep-wake transitions, and its dynamics across the day.

2) In Figure 4, they show that in the KO mice, that there are changes in expression of several clock genes and they imply they are all regulated by CIRBP. Since CLOCK and PER are transcriptional regulators, it is possible that only one is regulated by CIRBP and that the effects on other genes are indirect (through the clock). This should be discussed.

Thank you for this insightful feedback, and we apologize for not having sufficiently emphasized that some of the molecular changes might indeed not be direct but secondary to changes in CIRBP’s primary targets such as *Clock* and others. We now have specifically addressed this issue in the Discussion section ‘CIRBP adjusts clock gene expression and REM sleep recovery following sleep deprivation’.

The findings on clock gene regulation by SD are puzzling and in part contradict the results expected from the Schibler paper (Morf et al., 2012). This suggests a different mechanism of Cirbp-clock interaction in the cortex compared to fibroblasts. I think this is a critical point to understand the KO phenotype. Considering the differential responses of Clock and Rev-Erb, one could evaluate their contribution to SD responses by testing respective mouse mutants (and, possibly, double mutants).

Indeed, some aspects of our results do not follow our straightforward “sleep deprivation-temperature-CIRBP-clock genes” hypothesis that is central to our manuscript and which was based on the *in vitro* findings of Schibler and colleagues in fibroblasts published in Science. While the *Rev-Erbα* results do confirm our hypothesis, we agree that the increase in the sleep-deprivation induced changes in *Clock* and *Per2* expression in *Cirbp* KO mice is puzzling. In his recent review (Schibler et al., 2015), Ueli Schibler mentions unpublished data indicating that the amplitude of *in vivo* clock-gene rhythms in the liver of *Cirbp* KO mice was, surprisingly, increased (see legend of Figure 3 in that review). With the reviewer, he suggested that CIRBP’s effects on clock genes might be cell-type or tissue specific. In light of our results in the cortex, that go in a similar direction as those obtained in liver, it seems, however, that differences might be related to *in vitro* versus *in vivo* conditions instead. We have discussed this important issue in the section “Other mechanisms linking sleep-wake state to clock gene expression”.

“We […] hypothesized that some of the sleep deprivation-induced changes in clock gene expression occur through CIRBP” (quoted from the Introduction). In the past we have identified other pathways that contribute to the sleep-deprivation induced changes in clock-gene expression such as corticosterone signalling (Mongrain et al., 2010). Also lack of one of more clock genes affects the sleep-deprivation incurred changes in the expression of other clock genes (reviewed in Franken, 2013). As to testing *Clock* and *Rev-Erbα* mutant mice specifically, the absence of *Rev-Erbα*, a transcriptional repressor, led to the expected increase in the cortical expression of other clock genes (*Bmal1*, *Clock*, and *Npas2*) in *Rev-Erbα* KO mice. The cortical molecular response to sleep deprivation remained, however, unaffected (Mang et al., 2016). We did not quantify clock-gene expression in *Clock* KO mice but did so in KO mice for the perhaps more relevant (for the forebrain) *Clock*-paralog *Npas2* (Franken et al., 2006). We found that the sleep-deprivation induced increase in *Per2* expression in the brain of *Npas2* KO mice was significantly reduced. Although the latter finding demonstrates that the clock-gene circuitry can contribute to aspects of the sleep-deprivation induced changes in clock-gene expression, it has become evident that clock genes, especially *Per2*, can respond to a variety of systemic cues including stress, feeding, and, importantly, temperature, notably in the absence of a functional clock-gene circuitry (for reviews see Franken, 2013 and Schibler et al., 2015). With the CIRBP-temperature axis we have continued to investigate the influence of these systemic cues on the clock-gene circuitry and the collective data increasingly favour a scenario in which sleep-wake behaviour (through e.g. temperature, corticosterone, and feeding) drives rhythms is clock-gene expression in tissues peripheral to the SCN. We have addressed these issues in the section “Other mechanisms linking sleep-wake state to clock gene expression”.

3) Figure 6: Also, for locomotor activity, the specific contribution of the clock to Cirb-mediated effects should be tested.In Figure 6A, it would be interesting to phenotype these mice in a skeletal photoperiod. Since they are showing differences in sleep homeostasis, it is possible that activity is being 'masked' during the light phase. What is the intensity of light in these experiments?

Light was provided by overhead fluorescent tubes yielding an intensity of 70-90 lux at the cage-floor level. This information has now been further detailed in the Materials and methods.

In two cohorts of mice (WT n=5, KO n=7), we quantified circadian rhythms in locomotor activity under constant-dark (DD) conditions. The estimates of circadian tau mentioned in the manuscript (23.8h in both genotypes) were based on this sub-population of mice. We used this dataset to address the issue of masking by light by comparing, within the same individuals, mean levels of locomotor activity obtained in the rest phase under LD12:12 with the levels reached during the rest phase (or ‘rho’) under the ensuing DD (averages over 10 days in each lighting condition). This analysis did not reveal any evidence of a different response to light between genotypes (see Author response table 1). The sleep homeostatic phenotype we reported on concerned REM sleep recovery that was less efficient in KO mice. This REM sleep deficit was, however, most pronounced in the initial hours of the dark phase. For these reasons we do not think that this homeostatic phenotype can be attributed to (differences in) masking by light.

**Author response table 1**: Locomotor activity counts during the rest phase under LD12:12 and DD conditions (mean ± SEM)

**LD12:12****DD**
***Cirbp* KO**
34.0 ± 5.238.5 ± 4.2***Cirbp WT***35.5 ± 5.535.5 ± 5.9

2-way rANOVA w/ factors ‘Genotype’ (*P* = 0.91), ‘Light’ (*P* = 0.52), and their interaction (*P* = 0.52)